# MONOTONE NEAR-ZERO-SUM GAMES:
# A GENERALIZATION OF CONVEX-CONCAVE MINIMAX

**Ruichen Luo**[*]
IST Austria
3400 Klosterneuburg, Austria

**Sebastian U. Stich**
CISPA Helmholtz Center
66386 St. Ingbert, Germany

**Krishnendu Chatterjee**
IST Austria
3400 Klosterneuburg, Austria

## ABSTRACT

Zero-sum and non-zero-sum (aka general-sum) games are relevant in a wide range of applications. While general non-zero-sum games are computationally hard, researchers focus on the special class of monotone games for gradient-based algorithms. However, there is a substantial gap between the gradient complexity of monotone zero-sum and monotone general-sum games. Moreover, in many practical scenarios of games the zero-sum assumption needs to be relaxed. To address these issues, we define a new intermediate class of *monotone near-zero-sum games* that contains monotone zero-sum games as a special case. Then, we present a novel algorithm that transforms the near-zero-sum games into a sequence of zero-sum subproblems, improving the gradient-based complexity for the class. Finally, we demonstrate the applicability of this new class to model practical scenarios of games motivated from the literature.

## 1 INTRODUCTION

Two-player zero-sum games (also known as strictly competitive games) and their generalization to non-zero-sum games (Nash, 1951; Rosen, 1965) are crucial in domains like economics (von Neumann & Morgenstern, 1947), artificial intelligence (Yannakakis & Togelius, 2018), and biology in the form of evolutionary game theory (Weibull, 1997; Smith, 1982). As initiated in Rosen (1965); Tseng (1995); Nemirovski (2004), this paper focuses on the computationally tractable class of *monotone games* with compact convex strategy spaces.

Early work studies the restrictive classes of games where the two players have *equal conditioning*: the seminal paper of Nesterov (2005) established the complexity for bilinearly-coupled zero-sum games, which was later extended by Nemirovski (2004) to general-sum classes. Later on, the field explores the broader concept of *general conditioning*, which allows for more detailed portrayal of the two players (Chambolle & Pock, 2011; Lin et al., 2020). Early explorations of Chambolle & Pock (2011); Chen et al. (2014) focus on the bilinearly-coupled cases; starting from Lin et al. (2020), there has been extensive research into the generally-coupled, generally-conditioned, and zero-sum games (Yang et al., 2020; Wang & Li, 2020; Kovalev & Gasnikov, 2022; Lan & Li, 2023; Boţ et al., 2023). However, it has been an interesting open question whether these recent developments can be further generalized to the broader classes of generally-coupled, generally-conditioned, and *non-zero-sum* games.

On the theory side, there is a substantial gap between the gradient complexity of monotone zero-sum games and that of monotone general-sum games. The recent developments in minimax optimization (Lin et al., 2020; Kovalev & Gasnikov, 2022; Lan & Li, 2023) establish much better complexity results for the zero-sum cases compared to the long-standing bounds of general-sum games (Rockafellar, 1976; Tseng, 1995).

On the application side, strictly competitive scenarios, modeled by monotone zero-sum games, are often insufficient. Real-world game settings frequently involve factors such as transaction fees or semi-cooperation (Kalai & Kalai, 2013; Halpern & Rong, 2013), necessitating a relaxation of the zero-sum assumption in the modeling.

---

[*]This work was partially done during RL's stay at CISPA.

**Our contributions** Our work makes the first step towards bridging the gap between the monotone zero-sum and general-sum classes. For this purpose, we introduce a new intermediate class of monotone games, present a novel algorithm for this class, and show the applicability of this new class. In detail:

- *Theoretical motivation.* Our work extends the recent studies of monotone zero-sum games to the more general non-zero-sum settings. Specifically, we define a new intermediate class of games called *monotone near-zero-sum games*, characterized by a smoothness parameter $\delta$ describing the game's proximity to a zero-sum game. This new class of games presents a natural interpolation between monotone zero-sum games and a class of monotone general-sum games based on the near-zero-sum parameter $\delta$, and thus, it partially bridges the gap of the monotone zero-sum and general-sum classes.

- *Main theoretical result.* We propose a novel algorithm, Iterative Coupling Linearization (ICL), that provides a black-box reduction from monotone near-zero-sum games to zero-sum games. It converges to an $\varepsilon$-Nash equilibrium within $\widetilde{\mathcal{O}}\left(\left(\frac{L}{\sqrt{\mu\nu}} + \frac{L}{\min\{\mu,\nu\}} \cdot \min\left\{1, \sqrt{\frac{\delta}{\mu+\nu}}\right\}\right) \cdot \log^2\left(\frac{D^2}{\varepsilon}\right)\right)$ gradient queries,[1] where $L$ is the smoothness parameter, $\mu$ and $\nu$ are the strong concavity parameters of the two players, $\delta$ is the near-zero-sum parameter, and $D$ is the diameter. When $\delta$ is small, our results improve the long-standing complexity results of Tseng (1995); Nemirovski (2004) for the first time in non-zero-sum classes.

- *Practical applications.* Besides the theoretical motivation, we demonstrate the practical relevance of this new class of games. We consider *regularized matrix games* and *competitive games with small additional incentives*. These games are not zero-sum but naturally have a near-zero-sum structure, where our methods are applied to achieve provably faster rates.

## 2 DEFINITIONS, PREVIOUS RESULTS, AND THE NEW PROBLEM CLASS

### 2.1 BASIC DEFINITIONS

This paper studies the *Nash Equilibrium Problem* (NEP) for two-person general-sum games, in which Player 1 wants to maximize its utility function $u_1(\mathbf{x}, \mathbf{y})$ over $\mathbf{x} \in X$ and Player 2 wants to maximize its utility function $u_2(\mathbf{x}, \mathbf{y})$ over $\mathbf{y} \in Y$. Here, $X$ and $Y$ are compact and convex sets, and $u_1(\cdot, \cdot) \colon X \times Y \to \mathbb{R}$ and $u_2(\cdot, \cdot) \colon X \times Y \to \mathbb{R}$ are smooth functions. A pair of decisions $(\mathbf{x}^*, \mathbf{y}^*) \in X \times Y$ is a *Nash equilibrium* if

$$u_1(\mathbf{x}^*, \mathbf{y}^*) \geq u_1(\mathbf{x}, \mathbf{y}^*), \text{ for all } \mathbf{x} \in X, \text{ and } u_2(\mathbf{x}^*, \mathbf{y}^*) \geq u_2(\mathbf{x}^*, \mathbf{y}), \text{ for all } \mathbf{y} \in Y.$$

A pair of decisions $(\bar{\mathbf{x}}, \bar{\mathbf{y}}) \in X \times Y$ is an *$\varepsilon$-accurate Nash equilibrium* if there exists a Nash equilibrium $(\mathbf{x}^*, \mathbf{y}^*)$ such that $\|\bar{\mathbf{x}} - \mathbf{x}^*\|^2 + \|\bar{\mathbf{y}} - \mathbf{y}^*\|^2 \leq \varepsilon$. A pair of decisions $(\hat{\mathbf{x}}, \hat{\mathbf{y}}) \in X \times Y$ is an *$\varepsilon$-approximate Nash equilibrium*, if $u_1(\hat{\mathbf{x}}, \hat{\mathbf{y}}) \geq u_1(\mathbf{x}, \hat{\mathbf{y}}) - \varepsilon$ for all $\mathbf{x} \in X$ and $u_2(\hat{\mathbf{x}}, \hat{\mathbf{y}}) \geq u_2(\hat{\mathbf{x}}, \mathbf{y}) - \varepsilon$ for all $\mathbf{y} \in Y$. The relation between accurate and approximate Nash equilibria is discussed in Appendix B. The goal of this paper is to find an $\varepsilon$-accurate (or an $\varepsilon$-approximate) Nash equilibrium by iterative algorithms which subsequently query the gradients of the utility functions.

**Notations** Let $\mathcal{X}$ and $\mathcal{Y}$ be Euclidean spaces. In the space $\mathcal{X} \times \mathcal{Y}$, for all $\mathbf{z} = (\mathbf{x}, \mathbf{y}) \in \mathcal{X} \times \mathcal{Y}$ and $\mathbf{z}' = (\mathbf{x}', \mathbf{y}') \in \mathcal{X} \times \mathcal{Y}$, define $\langle \mathbf{z}', \mathbf{z}\rangle \stackrel{\text{def}}{=} \langle \mathbf{x}', \mathbf{x}\rangle + \langle \mathbf{y}', \mathbf{y}\rangle$. For all these spaces, the norms are those induced by inner products. Assume that the diameter of $X \subseteq \mathcal{X}$ is bounded by $D_X$ and the diameter of $Y \subseteq \mathcal{Y}$ is bounded by $D_Y$. Let $D = \sqrt{D_X^2 + D_Y^2}$. Assume that $u_1(\cdot, \cdot)$ and $u_2(\cdot, \cdot)$ are $L$-smooth, that is,

$$\|\nabla u_1(\mathbf{z}') - \nabla u_1(\mathbf{z})\| \leq L\|\mathbf{z}' - \mathbf{z}\|, \ \|\nabla u_2(\mathbf{z}') - \nabla u_2(\mathbf{z})\| \leq L\|\mathbf{z}' - \mathbf{z}\|, \text{ for all } \mathbf{z}, \mathbf{z}' \in X \times Y.$$

To facilitate our analysis, we adopt the following formulation that decomposes the game into a coupling part and a zero-sum part. Denote

$$g = -\frac{1}{2}(u_1 + u_2), \quad h = \frac{1}{2}(-u_1 + u_2), \quad \mathcal{H} = (\nabla_{\mathbf{x}}h, -\nabla_{\mathbf{y}}h), \quad \mathcal{F} = -(\nabla_{\mathbf{x}}u_1, \nabla_{\mathbf{y}}u_2),$$

---

[1] In the $\widetilde{\mathcal{O}}(\cdot)$ notations, the poly-logarithm terms are omitted.

where $g$ is the coupling part, $h$ is the zero-sum part, $\mathcal{H}$ is the operator corresponding to the zero-sum part $h$, and $\mathcal{F}$ is the operator corresponding to the game. Then, we have

$$u_1 = -g - h, \quad u_2 = -g + h, \quad \mathcal{F} = \nabla g + \mathcal{H}.$$

Since the utilities $u_1$ and $u_2$ are both $L$-smooth, we have the functions $g$ and $h$ are both $L$-smooth, and the operators $\mathcal{H}$ and $\mathcal{F}$ are both $L$-Lipschitz continuous. While similar decompositions can be found in the literature of variational inequalities and game theory (Nemirovski, 1995; Halpern & Rong, 2013; Chen et al., 2017; Hwang & Rey-Bellet, 2020), we emphasize that this notation is particularly suited to characterize our near-zero-sum games (to be defined later) for explicitly separating the non-zero-sum coupling part.

## 2.2 PROBLEM CLASSES

**Monotone general-sum games**  The seminal work of Nash (1951); Rosen (1965) establishes the existence of Nash equilibrium for concave games. But to obtain a tractable class of NEPs, further restrictions need to be considered (Rosen, 1965). Specifically, we make the following assumptions:

**Assumption 1** (Convex-concave zero-sum part). *There exists $(\mu, \nu) \in [0, L] \times [0, L]$ such that the function $h(\mathbf{x}, \mathbf{y}) - \frac{\mu}{2} \|\mathbf{x}\|^2$ is convex in $\mathbf{x}$ for any fixed $\mathbf{y} \in Y$, and the function $h(\mathbf{x}, \mathbf{y}) + \frac{\nu}{2} \|\mathbf{y}\|^2$ is concave in $\mathbf{y}$ for any fixed $\mathbf{x} \in X$.*

**Assumption 2** (jointly convex coupling part). *The function $g(\cdot, \cdot)$ is jointly convex.*

The operator $\mathcal{H} = (\nabla_{\mathbf{x}} h, -\nabla_{\mathbf{y}} h)$ is monotone with modulus $\min\{\mu, \nu\}$ under Assumption 1, and the operator $\nabla g$ is monotone under Assumption 2. Hence, under Assumptions 1 and 2, the game (or the operator $\mathcal{F} = \nabla g + \mathcal{H}$) is monotone with modulus $\min\{\mu, \nu\}$ (Rosen, 1965; Nemirovski, 1995), that is, $\langle \mathcal{F}(\mathbf{z}') - \mathcal{F}(\mathbf{z}), \mathbf{z}' - \mathbf{z} \rangle \geq \min\{\mu, \nu\} \cdot \|\mathbf{z}' - \mathbf{z}\|^2$, for all $\mathbf{z}, \mathbf{z}' \in X \times Y$.

In this paper, we refer to a game as a *monotone (general-sum) game* if it satisfies Assumptions 1 and 2, and refer to a game as a *strongly monotone* game if it is a monotone game with modulus $\mu, \nu > 0$. It is known that there exists a unique Nash equilibrium for strongly monotone games (Rosen, 1965).

**Monotone zero-sum games (convex-concave minimax optimization)**  We now consider a subclass: monotone zero-sum games. A two-person game is *zero-sum* if $g = 0$. A game is said to be a *monotone zero-sum game* if it is zero-sum and satisfies Assumption 1. Note that monotone zero-sum games trivially satisfy Assumption 2 (since $g = 0$ is convex), and therefore form a subclass of monotone general-sum games. By Sion's minimax theorem (Sion, 1958), the NEP for monotone zero-sum games is equivalent to *convex-concave minimax optimization*, that is, finding or approaching a saddle point of the function $h(\cdot, \cdot)$.

## 2.3 PREVIOUS RESULTS

We begin with a historical overview of the related study of NEPs for monotone games. Many prior studies focus on restrictive cases. Early seminal work by Nesterov and Nemirovski in the early 2000s primarily addressed the restrictive classes of games with equal conditioning ($\mu = \nu$). Nesterov (2005) initially studied the bilinearly-coupled, equally-conditioned, and zero-sum cases, which Nemirovski (2004) later generalized to cover *generally-coupled* and *general-sum* settings. Subsequently, research has shifted towards the broader concept of *general conditioning* ($\mu \neq \nu$). This shift is motivated by the need for more flexible modeling of player behaviors in realistic problems and can lead to substantially faster convergence rates, particularly when one of the player has a better conditioning (Chambolle & Pock, 2011; Lin et al., 2020). Early explorations focus on the bilinearly-coupled and zero-sum cases, developing various primal-dual algorithms in different oracle settings (Chambolle & Pock, 2011; Chen et al., 2014; Kolmogorov & Pock, 2021; Thekumparampil et al., 2022). More recently, the seminal work of Lin et al. (2020) spurred extensive research into *generally-coupled*, *generally-conditioned*, and zero-sum games (see Yang et al. (2020); Wang & Li (2020); Zhang et al. (2022); Kovalev & Gasnikov (2022); Boţ et al. (2023); Lan & Li (2023); Lin et al. (2025), among others).

Despite these advancements, we are not aware of any study addressing the more general settings of *generally-coupled*, *generally-conditioned*, and *non-zero-sum* games. For this particularly challenging class, the only established results are the long-standing bounds from Tseng (1995); Nemirovski

(2004). These bounds, however, remain a huge gap from the optimal rates achieved for the zero-sum cases (Lin et al., 2020; Kovalev & Gasnikov, 2022; Lan & Li, 2023).

Now, we formally outline the state-of-the-art gradient complexity results of monotone general-sum and zero-sum games within the general settings of *general couplings* and *general conditioning*. To simplify the presentation, we assume strong monotonicity for now in the Section 2.3, while the results for non-strongly monotone games are indeed similar (as we will discuss later). For general-sum games, the NEPs can be solved using variational inequality methods for the operator $\mathcal{F}$, leading to the following long-standing gradient complexity:

**Proposition 1** (Tseng (1995)). *For strongly monotone general-sum games, an $\varepsilon$-accurate Nash equilibrium can be found with the number of gradient queries bounded by* $\mathcal{O}\left(\frac{L}{\min\{\mu,\nu\}} \cdot \log\left(\frac{D^2}{\varepsilon}\right)\right)$ .

For the zero-sum cases, the gradient complexity can be significantly improved due to recent advances in minimax optimization.

**Proposition 2** (Lin et al. (2020); Kovalev & Gasnikov (2022); Zhang et al. (2022); Lan & Li (2023)). *For strongly monotone zero-sum games, an $\varepsilon$-accurate Nash equilibrium can be found with the number of gradient queries bounded by* $\mathcal{O}\left(\frac{L}{\sqrt{\mu\nu}} \cdot \log\left(\frac{D^2}{\varepsilon}\right)\right)$ . *This rate is minimax optimal, as* $\Omega\left(\frac{L}{\sqrt{\mu\nu}} \cdot \log\left(\frac{D^2}{\varepsilon}\right)\right)$ *gradient queries are required in general.*

### 2.4 The new problem class

**Theoretical motivation**  As shown in Propositions 1 and 2, a huge gap exists in the gradient complexities for solving NEPs for monotone general-sum games versus zero-sum games. This motivates the exploration of an intermediate problem class that partially bridges this gap.

**Monotone near-zero-sum games**  We introduce the class of *monotone $\delta$-near-zero-sum games*, which naturally interpolates between monotone zero-sum ($\delta = 0$)[2] and general-sum ($\delta = L$) games.

**Assumption 3** (Near-zero-sum). *Let $\delta \in [0, L]$ such that the function $g(\cdot, \cdot)$ is $\delta$-smooth.*

> **Definition** (MONOTONE NEAR-ZERO-SUM GAMES). *If a two-person general-sum game satisfies Assumptions 1 to 3, we call it a* monotone $\delta$-near-zero-sum game.

## 3 Algorithm and convergence analysis

We first focus on the algorithm for strongly monotone near-zero-sum games in Sections 3.1 and 3.2. Then, in Section 3.3, we present the results for (non-strongly) monotone near-zero-sum games.

### 3.1 Algorithm

For the zero-sum classes, Lin et al. (2020); Kovalev & Gasnikov (2022); Carmon et al. (2022); Lan & Li (2023) obtained the optimal convergence rate by the Catalyst method (Lin et al., 2018). However, for the more general classes of non-zero-sum games, the application of Catalyst is complicated by the fact that the regularized minimization transforms the problem into a Stackelberg game, whose solution deviates significantly from a Nash equilibrium (see Appendix C for more details). Thus, we are not aware of how the similar smoothing techniques can be applied directly to non-zero-sum games.

This raises the challenge: *can we leverage the off-the-shelf algorithms designed for zero-sum games to solve the non-zero-sum problems of interest?* Now, we introduce our novel algorithm, Iterative Coupling Linearization (ICL), which overcomes the aforementioned challenge and presents a clean black-box framework to solve near-zero-sum games by using zero-sum algorithms as an oracle.

---

[2]In 0-near-zero-sum game, let Player 1 maximize $\mathbf{a}_1^\top \mathbf{x} + \mathbf{b}_1^\top \mathbf{y} - h(\mathbf{x}, \mathbf{y})$ and Player 2 maximize $\mathbf{a}_2^\top \mathbf{x} + \mathbf{b}_2^\top \mathbf{y} + h(\mathbf{x}, \mathbf{y})$, respectively. The Nash equilibrium in the above game is the same as that in the following zero-sum game: Player 1 maximizes $\mathbf{a}_1^\top \mathbf{x} - \mathbf{b}_2^\top \mathbf{y} - h(\mathbf{x}, \mathbf{y})$ and Player 2 maximizes $-\mathbf{a}_1^\top \mathbf{x} + \mathbf{b}_2^\top \mathbf{y} + h(\mathbf{x}, \mathbf{y})$.

---

**Algorithm 1** Iterative Coupling Linearization (ICL)

---

**Require:** $\mathbf{z}_0 = (\mathbf{x}_0, \mathbf{y}_0) \in X \times Y$.

1: **for** $t = 0, 1, \cdots, T-1$ **do**

2:  Let $\varphi_t(\mathbf{x}, \mathbf{y}) \overset{\text{def}}{=}$

$$\langle \nabla_{\mathbf{x}} g(\mathbf{x}_t, \mathbf{y}_t), \mathbf{x} \rangle + \frac{1}{2\eta_t} \|\mathbf{x} - \mathbf{x}_t\|^2 + h(\mathbf{x}, \mathbf{y}) - \langle \nabla_{\mathbf{y}} g(\mathbf{x}_t, \mathbf{y}_t), \mathbf{y} \rangle - \frac{1}{2\eta_t} \|\mathbf{y} - \mathbf{y}_t\|^2 . \quad (1)$$

3:  Find an inexact solution $\mathbf{z}_{t+1} = (\mathbf{x}_{t+1}, \mathbf{y}_{t+1}) \in X \times Y$ to $\min_{\mathbf{x} \in X} \max_{\mathbf{y} \in Y} \varphi_t(\mathbf{x}, \mathbf{y})$ such that

$$\langle \nabla_{\mathbf{x}} \varphi_t(\mathbf{z}_{t+1}), \mathbf{x}_{t+1} - \mathbf{x} \rangle - \langle \nabla_{\mathbf{y}} \varphi_t(\mathbf{z}_{t+1}), \mathbf{y}_{t+1} - \mathbf{y} \rangle \leq \varepsilon_t, \text{ for all } \mathbf{x} \in X, \ \mathbf{y} \in Y . \quad (2)$$

4: **end for**

---

**Potential function** Our algorithm leverages a natural potential function $\Delta \colon X \times Y \to \mathbb{R}$ defined as:

$$\Delta(\mathbf{z}) = \max_{\widetilde{\mathbf{z}} = (\widetilde{\mathbf{x}}, \widetilde{\mathbf{y}}) \in X \times Y} \underbrace{g(\mathbf{z}) - g(\widetilde{\mathbf{z}})}_{\text{jointly convex coupling}} + \underbrace{h(\mathbf{x}, \widetilde{\mathbf{y}}) - h(\widetilde{\mathbf{x}}, \mathbf{y})}_{\text{convex-concave zero-sum}}, \text{ for all } \mathbf{z} = (\mathbf{x}, \mathbf{y}) \in X \times Y .$$

This potential function decomposes into a jointly convex coupling part and a convex-concave zero-sum part. We show below in Propositions 3 and 4 that minimizing this potential function $\Delta(\cdot)$ is sufficient for finding a Nash equilibrium (with detailed proofs in Appendix D.1):

**Proposition 3.** *For any* $\mathbf{z} = (\mathbf{x}, \mathbf{y}) \in X \times Y$, *we have* $\Delta(\mathbf{z}) \geq 0$ *and*

$$2\Delta(\mathbf{z}) \geq \max_{\widetilde{\mathbf{z}} = (\widetilde{\mathbf{x}}, \widetilde{\mathbf{y}}) \in X \times Y} u_1(\widetilde{\mathbf{x}}, \mathbf{y}) - u_1(\mathbf{x}, \mathbf{y}) + u_2(\mathbf{x}, \widetilde{\mathbf{y}}) - u_2(\mathbf{x}, \mathbf{y}) .$$

**Proposition 4.** *Let* $\mathbf{z}^* \in X \times Y$. *In monotone games,* $\mathbf{z}^*$ *is the Nash equilibrium iff* $\Delta(\mathbf{z}^*) = 0$.

**Algorithm description** Our ICL algorithm solves the strongly monotone near-zero-sum game by iteratively linearizing the coupling part, thereby transforming the non-zero-sum game into a sequence of strongly monotone zero-sum subproblems. The pseudocode is presented in Algorithm 1.

Specifically, at every iteration $t$, we linearize the coupling part in the potential function at $\mathbf{z}_t$:

$$\min_{\mathbf{z} \in X \times Y} \Delta(\mathbf{z}) \rightsquigarrow \min_{\mathbf{z} \in X \times Y} \max_{\widetilde{\mathbf{z}} \in X \times Y} \langle \nabla g(\mathbf{z}_t), \mathbf{z} - \widetilde{\mathbf{z}} \rangle + \frac{1}{2\eta_t} \left( \|\mathbf{z} - \mathbf{z}_t\|^2 - \|\widetilde{\mathbf{z}} - \mathbf{z}_t\|^2 \right) + h(\mathbf{x}, \widetilde{\mathbf{y}}) - h(\widetilde{\mathbf{x}}, \mathbf{y}),$$

and note that this minimax optimization can be fully decomposed into two separate problems:

$$\min_{\mathbf{x} \in X} \max_{\widetilde{\mathbf{y}} \in Y} \langle \nabla g(\mathbf{z}_t), (\mathbf{x}, -\widetilde{\mathbf{y}}) \rangle + h(\mathbf{x}, \widetilde{\mathbf{y}}) + \frac{1}{2\eta_t} \left( \|\mathbf{x} - \mathbf{x}_t\|^2 - \|\widetilde{\mathbf{y}} - \mathbf{y}_t\|^2 \right) = \varphi_t(\mathbf{x}, \widetilde{\mathbf{y}}), \text{ and}$$

$$\min_{\mathbf{y} \in Y} \max_{\widetilde{\mathbf{x}} \in X} \langle \nabla g(\mathbf{z}_t), (-\widetilde{\mathbf{x}}, \mathbf{y}) \rangle - h(\widetilde{\mathbf{x}}, \mathbf{y}) + \frac{1}{2\eta_t} \left( -\|\widetilde{\mathbf{x}} - \mathbf{x}_t\|^2 + \|\mathbf{y} - \mathbf{y}_t\|^2 \right) = -\varphi_t(\widetilde{\mathbf{x}}, \mathbf{y}).$$

Moreover, by Sion's minimax theorem (Sion, 1958), (after simple substitutions) these two separate problems unify into a *single* saddle point problem of $\min_{\mathbf{x} \in X} \max_{\mathbf{y} \in Y} \varphi_t(\mathbf{x}, \mathbf{y})$, where $\varphi_t$ is defined in Equation (1). The update $\mathbf{z}_{t+1} = (\mathbf{x}_{t+1}, \mathbf{y}_{t+1})$ is then computed by inexactly solving this *unified* saddle point problem, where the inexactness condition is specified in Equation (2). Our algorithm, thus, provides a clean and black-box reduction from near-zero-sum games to zero-sum games.

## 3.2 CONVERGENCE ANALYSIS

Throughout Section 3.2, let $\mathbf{z}^* = (\mathbf{x}^*, \mathbf{y}^*)$ be the (unique) Nash equilibrium for the game. We only present the main proof ideas in this section, and the detailed proofs can be found in Appendix D.2. The core of our convergence analysis is to use the properties of the potential function and derive the following descent lemma, based on which we have the outer loop convergence.

**Lemma 5** (Descent lemma). *In monotone $\delta$-near-zero-sum games, for $\eta_t \leq \frac{1}{\delta}$, Algorithm 1 ensures*

$$\left( \frac{1}{2\eta_t} + \frac{\min\{\mu, \nu\}}{2} \right) \|\mathbf{z}_{t+1} - \mathbf{z}^*\|^2 \leq \frac{1}{2\eta_t} \|\mathbf{z}_t - \mathbf{z}^*\|^2 + \varepsilon_t .$$

**Lemma 6** (Outer loop convergence). *Let $\eta_t = \eta \in (0, \frac{1}{\delta}]$, for all $t \in [0, T-1] \cap \mathbb{Z}$. Denote $\theta = \frac{\min\{\mu,\nu\}}{\eta^{-1}+\min\{\mu,\nu\}}$. Let $\varepsilon_t \leq \frac{\theta\varepsilon}{4\eta}$, for all $t \in [0, T-1] \cap \mathbb{Z}$. For strongly monotone $\delta$-near-zero-sum games, if the outer loop iterate $t \geq \frac{1}{\theta}\log\frac{2D^2}{\varepsilon}$, then Algorithm 1 converges to an $\varepsilon$-accurate Nash equilibrium, that is, $\|\mathbf{z}_t - \mathbf{z}^*\|^2 \leq \varepsilon$.*

For the inner loop, any optimal gradient method for zero-sum games can be used:

**Lemma 7** (Inner loop complexity (Kovalev & Gasnikov, 2022; Carmon et al., 2022; Thekumparampil et al., 2022; Lan & Li, 2023)). *Under Assumption 1 with modulus $\mu, \nu > 0$, at each iteration $t \in [0, T-1] \cap \mathbb{Z}$, for $\eta_t \geq \frac{1}{L}$, the inexact solution $(\mathbf{x}_{t+1}, \mathbf{y}_{t+1})$ in Equation (2) of Algorithm 1 can be found with the number of gradient queries bounded by $\mathcal{O}\left(\frac{L}{\sqrt{(\eta_t^{-1}+\mu)(\eta_t^{-1}+\nu)}} \cdot \log\left(\frac{LD^2}{\varepsilon_t}\right)\right)$.*

Combining the outer loop convergence result (Lemma 6) with the inner loop complexity (Lemma 7), we obtain the main theoretical result of this paper, the overall gradient complexity of Algorithm 1.

**Theorem 1** (Main theoretical result). *Denote $\eta = \min\left\{\frac{1}{\delta}, \frac{1}{\min\{\mu,\nu\}}\right\}$ and $\theta = \frac{\min\{\mu,\nu\}}{\eta^{-1}+\min\{\mu,\nu\}}$. Let $\eta_t = \eta$ and $\varepsilon_t = \frac{\theta\varepsilon}{4\eta}$, for all $t \in [0, T-1] \cap \mathbb{Z}$. For strongly monotone $\delta$-near-zero-sum games, for $T \geq \frac{1}{\theta}\log\frac{2D^2}{\varepsilon}$, the outer loop iterates of Algorithm 1 converge to an $\varepsilon$-accurate Nash equilibrium with the number of gradient queries bounded by*

$$\mathcal{O}\left(\left(\frac{L}{\sqrt{\mu\nu}} + \frac{L}{\min\{\mu,\nu\}} \cdot \min\left\{1, \sqrt{\frac{\delta}{\mu+\nu}}\right\}\right) \cdot \log\left(\frac{LD^2}{\min\{\mu,\nu\}\cdot\varepsilon}\right)\log\left(\frac{D^2}{\varepsilon}\right)\right).$$

Finally, we highlight the conditions under which Algorithm 1 achieves a faster convergence rate compared to variational inequality methods (Tseng, 1995).

REMARK 1 (Acceleration in strongly monotone near-zero-sum games). *Consider strongly monotone near-zero-sum games where $\min\{\mu,\nu\}+\delta = o\left(\max\{\mu,\nu\}\right)$. The gradient complexity of Algorithm 1 is $\widetilde{\mathcal{O}}\left(\left(\frac{L}{\sqrt{\mu\nu}} + \frac{L}{\min\{\mu,\nu\}} \cdot \sqrt{\frac{\delta}{\mu+\nu}}\right) \cdot \log^2\left(\frac{D^2}{\varepsilon}\right)\right)$, which (ignoring logarithms) improves upon the $\mathcal{O}\left(\frac{L}{\min\{\mu,\nu\}} \cdot \log\left(\frac{D^2}{\varepsilon}\right)\right)$ rate of variational inequality methods as stated in Proposition 1. We also remark that for the special case of zero-sum games ($\delta = 0$), our rate recovers the optimal $\mathcal{O}\left(\frac{L}{\sqrt{\mu\nu}} \cdot \log\left(\frac{D^2}{\varepsilon}\right)\right)$ rate as stated in Proposition 2 up to a logarithm term.*

### 3.3 ACCELERATION IN NON-STRONGLY MONOTONE NEAR-ZERO-SUM GAMES

We first state the known results for non-strongly monotone games in literature.

**Proposition 8** (Nemirovski (2004)). *In monotone general-sum games where $\mu = 0$ or $\nu = 0$, an $\varepsilon$-approximate Nash equilibrium can be found within $\mathcal{O}\left(\frac{LD^2}{\varepsilon}\right)$ gradient queries.*

Then, we provide our result, which is obtained by a similar reduction as in Lin et al. (2020); Wang & Li (2020); Thekumparampil et al. (2022). The proof can be found in Appendix D.3.

**Corollary 9.** *In monotone $\delta$-near-zero-sum games with modulus $\mu$ and $\nu$:*

*(a) For $\mu > 0$ and $\nu = 0$, an $\varepsilon$-approximate Nash equilibrium can be obtained within $\widetilde{\mathcal{O}}\left(\left(\frac{LD_Y}{\sqrt{\mu\varepsilon}} + \frac{LD_Y^2}{\varepsilon} \cdot \min\left\{1, \sqrt{\frac{\delta}{\mu}}\right\}\right) \cdot \log^2\left(\frac{LD^2}{\varepsilon}\right)\right)$ gradient queries;*

*(b) For $\mu = 0$ and $\nu > 0$, an $\varepsilon$-approximate Nash equilibrium can be obtained within $\widetilde{\mathcal{O}}\left(\left(\frac{LD_X}{\sqrt{\nu\varepsilon}} + \frac{LD_X^2}{\varepsilon} \cdot \min\left\{1, \sqrt{\frac{\delta}{\nu}}\right\}\right) \cdot \log^2\left(\frac{LD^2}{\varepsilon}\right)\right)$ gradient queries; and*

*(c) For $\mu = 0$ and $\nu = 0$, an $\varepsilon$-approximate Nash equilibrium can be obtained within $\widetilde{\mathcal{O}}\left(\left(\frac{LD_XD_Y}{\varepsilon} + \frac{LD^2}{\varepsilon} \cdot \min\left\{1, \sqrt{\frac{\delta}{\varepsilon/D_X^2+\varepsilon/D_Y^2}}\right\}\right) \cdot \log^2\left(\frac{LD^2}{\varepsilon}\right)\right)$ gradient queries.*

REMARK 2 (Acceleration in non-strongly monotone near-zero-sum games). *For non-strongly monotone general-sum games, our rate of finding an $\varepsilon$-approximate Nash equilibrium (ignoring logarithms) is faster than the $\mathcal{O}\left(\frac{LD^2}{\varepsilon}\right)$ rate in literature: (a) when $\delta = o(\mu)$ and $\nu = 0$; (b) when $\mu = 0$ and $\delta = o(\nu)$; or (c) when $\mu = \nu = 0$ and $\varepsilon/D^2 + \delta = o\left(\varepsilon/D_X^2 + \varepsilon/D_Y^2\right)$. We also remark that for the special case of zero-sum games ($\delta = 0$), our rates recover the optimal rates of Lin et al. (2020); Wang & Li (2020) up to logarithm terms.*

**Technical novelty** Our technical contributions include a novel coupling linearization technique and the derivation of a descent lemma (Lemma 5). These elements combine to form a clean, general, and powerful black-box reduction. This reduction's key advantage is its ability to solve monotone non-zero-sum problems by treating any off-the-shelf zero-sum algorithm as an oracle. Moreover, the inherent black-box nature of our method also allows for deriving complexity results for other oracle settings. For instance, we present some additional results with proximal oracles in Appendix E.

## 4 APPLICATION EXAMPLES

In this section, we present practical examples of monotone near-zero-sum games. We focus on the application of our approach, while the proof details are presented in Appendix D.4.

### 4.1 OUR APPROACH FOR REGULARIZED MATRIX GAMES

**Regularized matrix games** We demonstrate the applicability of Iterative Coupling Linearization to regularized matrix games. Let $X \subseteq \mathbb{R}^n$ and $Y \subseteq \mathbb{R}^m$ be compact convex sets, and let $\mathbf{A}, \mathbf{B} \in \mathbb{R}^{m \times n}$ with $\|\mathbf{A}\| \leq L$, $\|\mathbf{B}\| \leq L$, and $\left\|\frac{\mathbf{A}+\mathbf{B}}{2}\right\| \leq \beta$. Let $\mathcal{R} \colon X \times Y \to \mathbb{R}$ be an $L$-smooth regularizer that is $\mu$-strongly concave-$\nu$-strongly convex. Let Player 1 maximize

$$u_1(\mathbf{x}, \mathbf{y}) = \langle \mathbf{A}\mathbf{x}, \mathbf{y} \rangle + \mathcal{R}(\mathbf{x}, \mathbf{y})$$

over $\mathbf{x} \in X$ and Player 2 maximize

$$u_2(\mathbf{x}, \mathbf{y}) = \langle \mathbf{B}\mathbf{x}, \mathbf{y} \rangle - \mathcal{R}(\mathbf{x}, \mathbf{y})$$

over $\mathbf{y} \in Y$. Assume that $\beta \leq \frac{1}{2}\sqrt{\mu\nu}$, then the game is $\min\{\frac{\mu}{2}, \frac{\nu}{2}\}$-strongly monotone, so classic variational inequality methods yield an $\varepsilon$-accurate Nash equilibrium within $\mathcal{O}\left(\frac{L}{\min\{\mu,\nu\}} \cdot \log\left(\frac{D^2}{\varepsilon}\right)\right)$ gradient queries (Tseng, 1995).

Now, we show that our ICL algorithm can be applied to get a faster rate by leveraging the near-zero-sum structure. Since $-\frac{1}{2}(u_1 + u_2)(\cdot, \cdot)$ is not jointly convex, violating Assumption 2, we first apply a "convex reformulation" technique.

**Convex reformulation technique** Specifically, we choose the parameters $\beta_1$ and $\beta_2$ based on the relationship between $2\beta$, $\mu$, and $\nu$: (i) If $2\beta \leq \mu$ and $2\beta \leq \nu$, let $\beta_1 = \beta_2 = \beta$; (ii) if $\mu \leq 2\beta \leq \nu$, let $\beta_1 = \frac{\mu}{2}$ and $\beta_2 = \frac{2\beta^2}{\mu}$; and (iii) if $\nu \leq 2\beta \leq \mu$, let $\beta_1 = \frac{2\beta^2}{\nu}$ and $\beta_2 = \frac{\nu}{2}$. With these choices, we always have $\beta_1 \leq \frac{\mu}{2}$, $\beta_2 \leq \frac{\nu}{2}$, and $\sqrt{\beta_1\beta_2} = \beta$.

We then reformulate the problem as follows: Player 1 maximizes $\widetilde{u}_1(\mathbf{x}, \mathbf{y}) = u_1(\mathbf{x}, \mathbf{y}) - \beta_2 \|\mathbf{y}\|^2$ over $\mathbf{x} \in X$, and Player 2 maximizes $\widetilde{u}_2(\mathbf{x}, \mathbf{y}) = u_2(\mathbf{x}, \mathbf{y}) - \beta_1 \|\mathbf{x}\|^2$ over $\mathbf{y} \in Y$. This reformulated NEP has the same Nash equilibrium as the original. Let

$$\widetilde{g}(\mathbf{x}, \mathbf{y}) = -\left\langle \left(\frac{\mathbf{A} + \mathbf{B}}{2}\right)\mathbf{x}, \mathbf{y} \right\rangle + \left(\frac{\beta_1}{2}\|\mathbf{x}\|^2 + \frac{\beta_2}{2}\|\mathbf{y}\|^2\right)$$

and

$$\widetilde{h}(\mathbf{x}, \mathbf{y}) = -\left\langle \left(\frac{\mathbf{A} - \mathbf{B}}{2}\right)\mathbf{x}, \mathbf{y} \right\rangle - \left(\mathcal{R}(\mathbf{x}, \mathbf{y}) + \frac{\beta_1}{2}\|\mathbf{x}\|^2 - \frac{\beta_2}{2}\|\mathbf{y}\|^2\right).$$

Then, $\widetilde{u}_1 = -\widetilde{g} - \widetilde{h}$ and $\widetilde{u}_2 = -\widetilde{g} + \widetilde{h}$. Since $\beta \leq \frac{1}{2}\sqrt{\mu\nu}$, by Cauchy-Schwartz inequality, $\widetilde{g}(\cdot, \cdot)$ is jointly convex. Further, by the choices of $\beta_1$ and $\beta_2$, we have $\widetilde{g}(\cdot, \cdot)$ is $(\beta + \max\{\beta_1, \beta_2\})$-smooth, and $\widetilde{h}(\cdot, \cdot)$ is $\frac{\mu}{2}$-strongly convex-$\frac{\nu}{2}$-strongly concave.

**Our approach applied to reformulated games** Now applying Algorithm 1 to the reformulated NEP, by Theorem 1, we obtain an $\varepsilon$-accurate Nash equilibrium with the number of gradient queries bounded by

$$\widetilde{\mathcal{O}}\left(\left(\frac{L}{\sqrt{\mu\nu}} + \frac{L}{\min\{\mu,\nu\}} \cdot \frac{\beta}{\sqrt{\mu\nu}}\right) \cdot \log^2\left(\frac{D^2}{\varepsilon}\right)\right).$$

When $\min\{\mu,\nu\} + \beta = o\left(\frac{1}{2}\sqrt{\mu\nu}\right)$, this rate surpasses the best-known $\mathcal{O}\left(\frac{L}{\min\{\mu,\nu\}} \cdot \log\left(\frac{D^2}{\varepsilon}\right)\right)$ gradient complexity of variational inequality methods (Tseng, 1995). This acceleration leveraging the near-zero-sum structure seems to be a new result even in the well-studied context of matrix games.

EXAMPLE 1 (Matrix games with transaction fees). *Consider regularized matrix games with transaction fees. Let* $X = \mathcal{P}_n \overset{\text{def}}{=} \left\{\mathbf{x} \in \mathbb{R}^n_{\geq 0} \mid x_1 + \cdots + x_n = 1\right\}$ *and* $Y = \mathcal{P}_m \overset{\text{def}}{=} \left\{\mathbf{y} \in \mathbb{R}^m_{\geq 0} \mid y_1 + \cdots + y_m = 1\right\}$. *Let* $\mathbf{M} \in \mathbb{R}^{m \times n}$ *be the payoff matrix of Player 1* without transaction fee*, with* $-\mathbf{M}$ *as the payoff matrix of Player 2* without transaction fee*. Assume* $\|\mathbf{M}\| \leq L$. *Denote* $\mathbf{M}_+ \overset{\text{def}}{=} \frac{1}{2}(\mathbf{M} + \text{abs}(\mathbf{M}))$ *and* $\mathbf{M}_- \overset{\text{def}}{=} \frac{1}{2}(-\mathbf{M} + \text{abs}(\mathbf{M}))$.[3]

*Now, suppose there is a* transaction fee *of* $\rho \in [0, 1]$ *charged by some third party on every payment. Then, the payoff matrices of Player 1 and Player 2* with transaction fees *are*

$$\mathbf{A} = (1 - \rho)\mathbf{M}_+ - \mathbf{M}_- \quad and \quad \mathbf{B} = -\mathbf{M}_+ + (1 - \rho)\mathbf{M}_-.$$

*Let* $\mathcal{R}\colon X \times Y \to \mathbb{R}$ *be an* $L$-smooth regularizer that is $\mu$-strongly concave-$\nu$-strongly convex*. Assume that* $\rho\|\text{abs}(\mathbf{M})\| = o\left(\sqrt{\mu\nu}\right)$.

*Let Player 1 maximize* $u_1(\mathbf{x}, \mathbf{y}) = \langle\mathbf{A}\mathbf{x}, \mathbf{y}\rangle + \mathcal{R}(\mathbf{x}, \mathbf{y})$ *over* $\mathbf{x} \in X$, *and Player 2 maximize* $u_2(\mathbf{x}, \mathbf{y}) = \langle\mathbf{B}\mathbf{x}, \mathbf{y}\rangle - \mathcal{R}(\mathbf{x}, \mathbf{y})$ *over* $\mathbf{y} \in Y$. *Applying the convex reformulation and then Algorithm 1, we obtain an* $\varepsilon$-accurate Nash equilibrium with the number of gradient queries bounded by

$$\widetilde{\mathcal{O}}\left(\left(\frac{L}{\sqrt{\mu\nu}} + \frac{L}{\min\{\mu,\nu\}} \cdot \frac{\rho\|\text{abs}(\mathbf{M})\|}{\sqrt{\mu\nu}}\right) \cdot \log^2\left(\frac{D^2}{\varepsilon}\right)\right).$$

*Matrix games with transaction fees are analogous to many practical scenarios, for instance, (a) betting sites often set odds such that there is overall no loss-gain (zero-sum game) but charge a small transaction fee; or (b) negotiation between two clients via a trusted-third party that charges a brokerage fee is another example of a case with small transaction fee.*

### 4.2 OUR APPROACH FOR COMPETITIVE GAMES WITH SMALL ADDITIONAL INCENTIVES

We show the applicability of Iterative Coupling Linearization to competitive games with small additional incentives. Let $X$ and $Y$ be compact convex sets in Euclidean spaces. Let $h\colon X \times Y \to \mathbb{R}$ be the competition payoff function, which is $L$-smooth and $\mu$-strongly convex-$\nu$-strongly concave. Let $g\colon X \times Y \to \mathbb{R}$ be the additional incentive function, which is $\beta$-smooth with $\beta \leq L$. Let Player 1 maximize $u_1 = -g - h$ over $\mathbf{x} \in X$, and Player 2 maximize $u_2 = -g + h$ over $\mathbf{y} \in Y$.

We explore two scenarios where the games are $\min\{\frac{\mu}{2}, \frac{\nu}{2}\}$-strongly monotone, to which our ICL algorithm as well as the classic variational inequalities (Tseng, 1995) can be applied:

1. If $g(\cdot, \cdot)$ is jointly convex and $\beta = o\left(\max\{\mu, \nu\}\right)$, applying Algorithm 1 directly yields an $\varepsilon$-accurate Nash equilibrium with the number of gradient queries bounded by

$$\widetilde{\mathcal{O}}\left(\left(\frac{L}{\sqrt{\mu\nu}} + \frac{L}{\min\{\mu,\nu\}} \cdot \sqrt{\frac{\beta}{\mu + \nu}}\right) \cdot \log^2\left(\frac{D^2}{\varepsilon}\right)\right).$$

2. If $\beta = o\left(\frac{1}{2}\min\{\mu, \nu\}\right)$, we first apply the "convex reformulation" technique. We reformulate the problem as follows: Player 1 maximizes $\widetilde{u}_1(\mathbf{x}, \mathbf{y}) = u_1(\mathbf{x}, \mathbf{y}) - \beta\|\mathbf{y}\|^2$ over $\mathbf{x} \in X$, and Player 2 maximizes $\widetilde{u}_2(\mathbf{x}, \mathbf{y}) = u_2(\mathbf{x}, \mathbf{y}) - \beta\|\mathbf{x}\|^2$ over $\mathbf{y} \in Y$. This reformulated NEP has the same Nash equilibrium as the original. Let $\widetilde{g} = -\frac{1}{2}(\widetilde{u}_1 + \widetilde{u}_2)$ and $\widetilde{h} = \frac{1}{2}(-\widetilde{u}_1 + \widetilde{u}_2)$. Then, $\widetilde{h}(\cdot, \cdot)$ is $\frac{\mu}{2}$-strongly convex-$\frac{\nu}{2}$-strongly concave, and $\widetilde{g}(\cdot, \cdot)$ is jointly convex and $2\beta$-smooth.

---

[3] abs $(\mathbf{M})$ represents a matrix of the same dimensions as $\mathbf{M}$ where each element is the absolute value of the corresponding element in $\mathbf{M}$. A more concrete illustration is given in Appendix F.

Applying Algorithm 1 to the reformulated NEP, we obtain an $\varepsilon$-accurate Nash equilibrium with the number of gradient queries bounded by

$$\widetilde{\mathcal{O}}\left(\frac{L}{\sqrt{\mu\nu}}\cdot\log^2\left(\frac{D^2}{\varepsilon}\right)\right).$$

In both scenarios 1 and 2, our gradient queries are fewer than the $\mathcal{O}\left(\frac{L}{\min\{\mu,\nu\}}\cdot\log\left(\frac{D^2}{\varepsilon}\right)\right)$ gradient queries of the classic variational inequality methods (Tseng, 1995).

EXAMPLE 2 (Competitive games with small cooperation incentives). *Consider the games where cooperation coexists with competition. Let $X \subseteq X_a \times X_b$ and $Y \subseteq Y_a \times Y_b$ be compact convex sets in Euclidean spaces. For $\mathbf{x} = (\mathbf{x}_a, \mathbf{x}_b) \in X$, $\mathbf{x}_a \in X_a$ represents Player 1's effort in cooperation, and $\mathbf{x}_b \in X_b$ represents Player 1's effort in competition (and similarly for $\mathbf{y} = (\mathbf{y}_a, \mathbf{y}_b) \in Y$). Let $f_a\colon X_a \times Y_a \to \mathbb{R}$ be the cooperation incentive function given by*

$$f_a(\mathbf{x}_a, \mathbf{y}_a) = \mathcal{R}_1(\mathbf{x}_a) + \widetilde{g}(\mathbf{x}_a, \mathbf{y}_a) + \mathcal{R}_2(\mathbf{y}_a),$$

*where the regularizer $\mathcal{R}_1\colon X_a \to \mathbb{R}$ is $\mu$-strongly convex and $L$-smooth, the function $\widetilde{g}\colon X_a \times Y_a \to \mathbb{R}$ is jointly convex and $\beta$-smooth, and the regularizer $\mathcal{R}_2\colon Y_a \to \mathbb{R}$ is $\nu$-strongly convex and $L$-smooth. Let $f_b\colon X_b \times Y_b \to \mathbb{R}$ be the competition payoff function, which is $L$-smooth and $\mu$-strongly convex-$\nu$-strongly concave. Assume that $\beta = o\left(\max\{\mu, \nu\}\right)$.*

*Let Player 1 maximize $u_1(\mathbf{x}, \mathbf{y}) = -f_a(\mathbf{x}_a, \mathbf{y}_a) - f_b(\mathbf{x}_b, \mathbf{y}_b)$ over $\mathbf{x} \in X$, and Player 2 maximize $u_2(\mathbf{x}, \mathbf{y}) = -f_a(\mathbf{x}_a, \mathbf{y}_a) + f_b(\mathbf{x}_b, \mathbf{y}_b)$ over $\mathbf{y} \in Y$. Denoting $\widetilde{h}(\mathbf{x}, \mathbf{y}) = \mathcal{R}_1(\mathbf{x}_a) + f_b(\mathbf{x}_b, \mathbf{y}_b) - \mathcal{R}_2(\mathbf{y}_a)$, the NEP can be reformulated as Player 1 maximizing $\widetilde{u}_1(\mathbf{x}, \mathbf{y}) = -\widetilde{g}(\mathbf{x}_a, \mathbf{y}_a) - \widetilde{h}(\mathbf{x}, \mathbf{y})$ over $\mathbf{x} \in X$, and Player 2 maximizing $\widetilde{u}_2(\mathbf{x}, \mathbf{y}) = -\widetilde{g}(\mathbf{x}_a, \mathbf{y}_a) + \widetilde{h}(\mathbf{x}, \mathbf{y})$ over $\mathbf{y} \in Y$. Applying Algorithm 1 as detailed above, we obtain an $\varepsilon$-accurate Nash equilibrium with the number of gradient queries bounded by*

$$\widetilde{\mathcal{O}}\left(\left(\frac{L}{\sqrt{\mu\nu}} + \frac{L}{\min\{\mu,\nu\}}\cdot\sqrt{\frac{\beta}{\mu+\nu}}\right)\cdot\log^2\left(\frac{D^2}{\varepsilon}\right)\right).$$

*As a final remark, the modeling of the coexistence of competition and cooperation has been well-researched (for instance, see the studies of Nash (1950; 1953); Selten (1960); Raiffa (1952); Kalai & Rosenthal (1978); Kalai & Kalai (2013); Halpern & Rong (2013) on semi-cooperative games). Indeed, these theories are often applied to the scenarios where cooperation dominates, and optimization techniques have been used to accelerate the dominant cooperation part (Chen et al., 2017). Our work contributes to this line of research on semi-cooperative games where competition dominates, yet there is a small cooperation incentive. For example, repeated Prisoners' dilemma with a stochastic number of rounds with small benefit-to-cost ratio is an example of competition with small cooperation incentive (as benefit-to-cost ratio $b/c$ is small) (Nowak, 2006a;b; Sigmund, 2010). We give a more concrete illustration of donation games in Appendix F.*

## 5 BASIC NUMERICAL EXPERIMENTS

We conducted basic numerical experiments to validate our theoretical results, focusing on matrix games with transaction fees as in Example 1. We set $n = m = 10000$, $\mu = 10^{-4}$, $\nu = 1$, and $\varepsilon = 10^{-7}$. A sparse, random matrix $\mathbf{M} \in \mathbb{R}^{m \times n}$ such that $\|\mathbf{M}\| = 1$ was generated. The regularizer was defined as $\mathcal{R}(\mathbf{x}, \mathbf{y}) = -\frac{\mu}{2}\|\mathbf{x}\|^2 + \frac{\nu}{2}\|\mathbf{y}\|^2$. We varied the transaction fee $\rho$ from $\{0.00\%, 0.03\%, \cdots, 0.18\%\}$. Our implementation of ICL (Algorithm 1), detailed in Section 4.1, used the Lifted Primal-Dual method (Thekumparampil et al., 2022) for the inner loop. We compared ICL against the Optimistic Gradient Descent Ascent (OGDA) (Popov, 1980) and Extra-Gradient (EG) (Korpelevich, 1976) methods for variational inequalities. More details and additional experiments are provided in Appendix G, and our code can be found here: https://github.com/riekenluo/Monotone_Near_Zero_Sum_Games.

The numerical results, summarized in Table 1, demonstrate that: (i) our ICL algorithm converges faster when the game is closer to a zero-sum one; (ii) the classic variational inequality methods (EG or OGDA) do not benefit from the near-zero-sum structure; and (iii) our ICL algorithm is faster than the classic variational inequality methods when the game is sufficiently near-zero-sum. In particular,

Table 1: Gradient query counts (in thousands) to converge to an $\varepsilon$-accurate Nash equilibrium under various transaction fees. Error bars indicate 2-sigma variations across 10 independent runs.

| Transaction Fee $\rho$  Methods | 0.00% | 0.03% | 0.06% | 0.09% | 0.12% | 0.15% | 0.18% |
|---|---|---|---|---|---|---|---|
| ICL (Algorithm 1) | $\mathbf{9.1 \pm 0.0}$ | $\mathbf{22.6 \pm 0.4}$ | $\mathbf{42.2 \pm 0.3}$ | $\mathbf{65.0 \pm 0.3}$ | $\mathbf{75.7 \pm 0.3}$ | $113.7 \pm 0.7$ | $123.8 \pm 0.6$ |
| OGDA (Popov, 1980) | $93.9 \pm 0.5$ | $93.9 \pm 0.5$ | $93.9 \pm 0.5$ | $93.9 \pm 0.5$ | $93.9 \pm 0.5$ | $\mathbf{94.0 \pm 0.6}$ | $\mathbf{94.0 \pm 0.6}$ |
| EG (Korpelevich, 1976) | $132.9 \pm 0.8$ | $132.9 \pm 0.8$ | $132.9 \pm 0.8$ | $132.9 \pm 0.8$ | $132.9 \pm 0.8$ | $132.9 \pm 0.8$ | $132.9 \pm 0.8$ |

in the experiments, ICL requires fewer gradient queries to converge to an $\varepsilon$-accurate Nash equilibrium when the transaction fee $\rho \leq 0.12\%$. This empirical observation aligns with our theoretical prediction in Example 1, which suggests that ICL converges faster when $\rho \left\| \mathrm{abs}\left(\mathbf{M}\right) \right\| \ll \sqrt{\mu\nu} = 1\%$.

# 6 CONCLUSIONS, LIMITATIONS, AND FUTURE WORK

In this work we consider the class of monotone games and present a condition that naturally interpolates between the zero-sum and a non-zero-sum class. We develop an efficient gradient-based approach and show its applicability with several examples motivated from the literature.

There are some limitations of our work: (a) in our complexity there is a $\log^2(\frac{D^2}{\varepsilon})$ dependency rather than a single logarithm dependency, and whether this double logarithm dependency can be removed is an interesting question; and (b) whether lower-bound results can be obtained for the new class also remains a challenging question and may involve difficult construction of non-quadratic functions, especially given that the lower bound for the general-sum classes remains widely open relative to the long-standing upper bound established in Tseng (1995); Nemirovski (2004).

The current work focuses on two-player monotone non-zero-sum games, which constitutes a crucial first step in generalizing two-player monotone zero-sum games with general conditioning. A natural, though highly non-trivial, extension involves generalizing our fast convergence rates to the multiplayer near-zero-sum setting. We notice that, as a preliminary step, the fast rate in zero-sum games (Proposition 2) have not yet been extended to multiplayer settings. Indeed, this preliminary step is conceptually challenging, as the zero-sum condition in multiplayer games does not imply the strict competition found in two-player games. Furthermore, classic reductions in von Neumann & Morgenstern (1947) show that an $n$-player general-sum game can be reduced to an $(n + 1)$-player zero-sum game by letting the $(n+1)$th player take the negative of the summation of the first $n$ players. This implies that three-player zero-sum games are inherently no easier than two-player general-sum games. These results suggest that achieving a fast convergence rate in the multiplayer setting will likely require significantly stronger structural assumptions than the simple summation of utilities to zero. Hence, we leave the full generalization to the multiplayer setting for future research.

In addition to the above theoretical aspects, there are several other interesting directions as well: for example, (a) exploring other applications of regularized matrix games with near-zero-sum payoff matrices is an interesting direction; and (b) in the research of semi-cooperative games where competition dominates, applying our methods in more practical examples is another fruitful direction for future research.

## ACKNOWLEDGMENTS

The authors thank for the helpful discussion with Anton Rodomanov during the initial preparation of this project.

RL and KC acknowledge the support of ERC CoG 863818 (ForM-SMArt) and Austrian Science Fund (FWF) 10.55776/COE12.

**Ethics statement**

This is a theoretical research without any ethics concern. All the experiments are based on randomly generated data. The only LLM usage is to aid or polish writing.

**Reproducibility statement**

We include the detailed proofs of all the results in the Appendix. We include the source codes of all the experiments, including the scripts for the generation of random data. The codes are based on standard Python libraries (like `numpy` or `scipy`) and are easy to run.

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

## A    RELATED WORK

We discuss other classes of games in the literature that bridge the gap between zero-sum and general-sum games.

**Near network zero-sum games**   Near network zero-sum games (Hussain et al., 2023) define a class of games that is close to network zero-sum games in terms of maximum pairwise difference (Candogan et al., 2013; Hussain et al., 2023). Limited to the setting of two-person games, monotone near-zero-sum games considered in this paper differ in three aspects: (i) The utility functions in this paper can be general functions, rather than bilinear functions; (ii) the difference between near-zero-sum games and zero-sum games in this paper is characterized by (higher-order) smoothness parameter, rather than by function values; and (iii) the solution of near network zero-sum games is taken directly from the zero-sum case, which only guarantees convergence to a neighborhood of the Nash equilibrium.

**Rank-$k$ games**   In the setting of matrix games, one of the most significant attempts on bridging the gap between zero-sum and non-zero-sum games is the class of Rank-$k$ games introduced in Kannan & Theobald (2010). As a generalization of zero-sum matrix games, Kannan & Theobald (2010) study matrix games where $\text{rank}(\mathbf{A} + \mathbf{B}) = k$, where $\mathbf{A}$ and $\mathbf{B}$ are the payoff matrices of the two players. To find an approximate Nash equilibrium, an FPTAS exists when $k$ is small (Kannan & Theobald, 2010); to find an exact Nash equilibrium, Rank-1 games can be solved in polynomial time (Adsul et al., 2021), while Rank-3 games are already PPAD-hard (Mehta, 2018). It is crucial to emphasize that monotone near-zero-sum games, as considered in this paper, are fundamentally distinct from Rank-$k$ games. Specifically: (i) The utility functions in this paper can be general functions, rather than bilinear functions; (ii) matrix games can be sufficiently near-zero-sum but still have full rank; and (iii) the focus of this paper is on gradient-based algorithms and complexity within the Nemirovsky-Yudin optimization model (Nemirovsky & Yudin, 1983), while the study of Rank-$k$ games focuses on algorithms and complexity on Turing machines.

## B    RELATIONS TO APPROXIMATE NASH EQUILIBRIUM

Indeed, an approximate Nash equilibrium can be obtained from an accurate Nash equilibrium (Nemirovski, 2004). Below, we include this result for self-consistency.

**Proposition 10** (Nemirovski (2004))**.** *In a monotone general-sum game, let* $\mathbf{z}^* = (\mathbf{x}^*, \mathbf{y}^*)$ *be the Nash equilibrium. Let* $\bar{\mathbf{z}} = (\bar{\mathbf{x}}, \bar{\mathbf{y}}) \in X \times Y$ *and* $\gamma \in (0, \frac{1}{\sqrt{2}L}]$. *We have*

$$\max_{\mathbf{x} \in X, \, \mathbf{y} \in Y} u_1(\mathbf{x}, \hat{\mathbf{y}}) - u_1(\hat{\mathbf{x}}, \hat{\mathbf{y}}) + u_2(\hat{\mathbf{x}}, \mathbf{y}) - u_2(\hat{\mathbf{x}}, \hat{\mathbf{y}})$$
$$\leq \max_{\mathbf{x} \in X, \, \mathbf{y} \in Y} \langle \mathcal{F}(\hat{\mathbf{x}}, \hat{\mathbf{y}}), (\hat{\mathbf{x}}, \hat{\mathbf{y}}) - (\mathbf{x}, \mathbf{y}) \rangle$$
$$\leq \frac{2}{\gamma} \sqrt{D_X^2 + D_Y^2} \, \|\bar{\mathbf{z}} - \mathbf{z}^*\| \,,$$

*where* $\hat{\mathbf{x}} \overset{\text{def}}{=} \Pi_X (\bar{\mathbf{x}} + \gamma \nabla_{\mathbf{x}} u_1(\bar{\mathbf{x}}, \bar{\mathbf{y}}))$ *and* $\hat{\mathbf{y}} \overset{\text{def}}{=} \Pi_Y (\bar{\mathbf{y}} + \gamma \nabla_{\mathbf{y}} u_2(\bar{\mathbf{x}}, \bar{\mathbf{y}}))$.[4]

*Proof.* Denote $\mathbf{x}_+ \overset{\text{def}}{=} \Pi_X (\bar{\mathbf{x}} + \gamma \nabla_{\mathbf{x}} u_1(\hat{\mathbf{x}}, \hat{\mathbf{y}}))$, $\mathbf{y}_+ \overset{\text{def}}{=} \Pi_Y (\bar{\mathbf{y}} + \gamma \nabla_{\mathbf{y}} u_2(\hat{\mathbf{x}}, \hat{\mathbf{y}}))$, and $\mathbf{z}_+ \overset{\text{def}}{=} (\mathbf{x}_+, \mathbf{y}_+)$. Consider any $\widetilde{\mathbf{x}} \in X$ and $\widetilde{\mathbf{y}} \in Y$. By the assignment of $\hat{\mathbf{x}}$, we have

$$\langle \nabla_{\mathbf{x}} u_1(\bar{\mathbf{x}}, \bar{\mathbf{y}}), \hat{\mathbf{x}} \rangle - \frac{1}{2\gamma} \|\hat{\mathbf{x}} - \bar{\mathbf{x}}\|^2 \geq \langle \nabla_{\mathbf{x}} u_1(\bar{\mathbf{x}}, \bar{\mathbf{y}}), \mathbf{x}_+ \rangle - \frac{1}{2\gamma} \|\mathbf{x}_+ - \bar{\mathbf{x}}\|^2 + \frac{1}{2\gamma} \|\mathbf{x}_+ - \hat{\mathbf{x}}\|^2 \,. \quad (3)$$

By the assignment of $\mathbf{x}_+$, we have

$$\langle \nabla_{\mathbf{x}} u_1(\hat{\mathbf{x}}, \hat{\mathbf{y}}), \mathbf{x}_+ \rangle - \frac{1}{2\gamma} \|\mathbf{x}_+ - \bar{\mathbf{x}}\|^2 \geq \langle \nabla_{\mathbf{x}} u_1(\hat{\mathbf{x}}, \hat{\mathbf{y}}), \widetilde{\mathbf{x}} \rangle - \frac{1}{2\gamma} \|\bar{\mathbf{x}} - \widetilde{\mathbf{x}}\|^2 + \frac{1}{2\gamma} \|\mathbf{x}_+ - \widetilde{\mathbf{x}}\|^2 \,. \quad (4)$$

---

[4]In a Euclidean space $\mathcal{Q}$, for a non-empty, closed, and convex set $Q \subseteq \mathcal{Q}$ and a vertex $\mathbf{u} \in \mathcal{Q}$, let $\Pi_Q(\mathbf{u})$ denote the projection of $\mathbf{u}$ onto $Q$, that is, $\Pi_Q(\mathbf{u}) \overset{\text{def}}{=} \arg\min_{\mathbf{v} \in Q} \|\mathbf{u} - \mathbf{v}\|$.

In view of

$$\langle \nabla_{\mathbf{x}} u_1(\hat{\mathbf{x}}, \hat{\mathbf{y}}), \widetilde{\mathbf{x}} - \hat{\mathbf{x}} \rangle$$

$$= \langle \nabla_{\mathbf{x}} u_1(\hat{\mathbf{x}}, \hat{\mathbf{y}}), \mathbf{x}_+ - \hat{\mathbf{x}} \rangle + \langle \nabla_{\mathbf{x}} u_1(\hat{\mathbf{x}}, \hat{\mathbf{y}}), \widetilde{\mathbf{x}} - \mathbf{x}_+ \rangle$$

$$\overset{(4)}{\leq} \langle \nabla_{\mathbf{x}} u_1(\hat{\mathbf{x}}, \hat{\mathbf{y}}), \mathbf{x}_+ - \hat{\mathbf{x}} \rangle - \frac{1}{2\gamma} \|\mathbf{x}_+ - \bar{\mathbf{x}}\|^2 + \frac{1}{2\gamma} \|\bar{\mathbf{x}} - \widetilde{\mathbf{x}}\|^2 - \frac{1}{2\gamma} \|\mathbf{x}_+ - \widetilde{\mathbf{x}}\|^2$$

$$= \langle \nabla_{\mathbf{x}} u_1(\hat{\mathbf{x}}, \hat{\mathbf{y}}) - \nabla_{\mathbf{x}} u_1(\bar{\mathbf{x}}, \bar{\mathbf{y}}), \mathbf{x}_+ - \hat{\mathbf{x}} \rangle + \langle \nabla_{\mathbf{x}} u_1(\bar{\mathbf{x}}, \bar{\mathbf{y}}), \mathbf{x}_+ - \hat{\mathbf{x}} \rangle - \frac{1}{2\gamma} \|\mathbf{x}_+ - \bar{\mathbf{x}}\|^2$$

$$+ \frac{1}{2\gamma} \|\bar{\mathbf{x}} - \widetilde{\mathbf{x}}\|^2 - \frac{1}{2\gamma} \|\mathbf{x}_+ - \widetilde{\mathbf{x}}\|^2$$

$$\overset{(3)}{\leq} \langle \nabla_{\mathbf{x}} u_1(\hat{\mathbf{x}}, \hat{\mathbf{y}}) - \nabla_{\mathbf{x}} u_1(\bar{\mathbf{x}}, \bar{\mathbf{y}}), \mathbf{x}_+ - \hat{\mathbf{x}} \rangle - \frac{1}{2\gamma} \|\hat{\mathbf{x}} - \bar{\mathbf{x}}\|^2 - \frac{1}{2\gamma} \|\mathbf{x}_+ - \hat{\mathbf{x}}\|^2 \qquad (5)$$

$$+ \frac{1}{2\gamma} \|\bar{\mathbf{x}} - \widetilde{\mathbf{x}}\|^2 - \frac{1}{2\gamma} \|\mathbf{x}_+ - \widetilde{\mathbf{x}}\|^2$$

$$\leq L \|(\hat{\mathbf{x}}, \hat{\mathbf{y}}) - \bar{\mathbf{z}}\| \cdot \|\mathbf{x}_+ - \hat{\mathbf{x}}\| - \frac{1}{2\gamma} \|\hat{\mathbf{x}} - \bar{\mathbf{x}}\|^2 - \frac{1}{2\gamma} \|\mathbf{x}_+ - \hat{\mathbf{x}}\|^2$$

$$+ \frac{1}{2\gamma} \|\bar{\mathbf{x}} - \widetilde{\mathbf{x}}\|^2 - \frac{1}{2\gamma} \|\mathbf{x}_+ - \widetilde{\mathbf{x}}\|^2$$

$$\leq \frac{L}{2\sqrt{2}} \|(\hat{\mathbf{x}}, \hat{\mathbf{y}}) - \bar{\mathbf{z}}\|^2 - \frac{1}{2\gamma} \|\hat{\mathbf{x}} - \bar{\mathbf{x}}\|^2 + \frac{1}{2\gamma} \|\bar{\mathbf{x}} - \widetilde{\mathbf{x}}\|^2 - \frac{1}{2\gamma} \|\mathbf{x}_+ - \widetilde{\mathbf{x}}\|^2$$

(where we have used $\gamma \leq \frac{1}{\sqrt{2}L}$ in the last inequality), and similarly,

$$\langle \nabla_{\mathbf{y}} u_2(\hat{\mathbf{x}}, \hat{\mathbf{y}}), \widetilde{\mathbf{y}} - \hat{\mathbf{y}} \rangle \leq \frac{L}{2\sqrt{2}} \|(\hat{\mathbf{x}}, \hat{\mathbf{y}}) - \bar{\mathbf{z}}\|^2 - \frac{1}{2\gamma} \|\hat{\mathbf{y}} - \bar{\mathbf{y}}\|^2 + \frac{1}{2\gamma} \|\bar{\mathbf{y}} - \widetilde{\mathbf{y}}\|^2 - \frac{1}{2\gamma} \|\mathbf{y}_+ - \widetilde{\mathbf{y}}\|^2 , \quad (6)$$

we have

$$\langle \mathcal{F}(\hat{\mathbf{x}}, \hat{\mathbf{y}}), (\hat{\mathbf{x}}, \hat{\mathbf{y}}) - (\widetilde{\mathbf{x}}, \widetilde{\mathbf{y}}) \rangle$$

$$\overset{(5)(6)}{=} (\frac{L}{\sqrt{2}} - \frac{1}{2\gamma}) \|(\hat{\mathbf{x}}, \hat{\mathbf{y}}) - \bar{\mathbf{z}}\|^2 + \frac{1}{2\gamma} \|\bar{\mathbf{z}} - (\widetilde{\mathbf{x}}, \widetilde{\mathbf{y}})\|^2 - \frac{1}{2\gamma} \|\mathbf{z}_+ - (\widetilde{\mathbf{x}}, \widetilde{\mathbf{y}})\|^2 \qquad (7)$$

$$\leq \frac{1}{2\gamma} \|\bar{\mathbf{z}} - (\widetilde{\mathbf{x}}, \widetilde{\mathbf{y}})\|^2 - \frac{1}{2\gamma} \|\mathbf{z}_+ - (\widetilde{\mathbf{x}}, \widetilde{\mathbf{y}})\|^2 ,$$

where we have used $\gamma \leq \frac{1}{\sqrt{2}L}$ in the last inequality.

Taking $(\widetilde{\mathbf{x}}, \widetilde{\mathbf{y}}) := (\mathbf{x}^*, \mathbf{y}^*)$ in Equation (7) for the moment, we get

$$\|\mathbf{z}_+ - \mathbf{z}^*\|^2 \overset{(7)}{\leq} \|\bar{\mathbf{z}} - \mathbf{z}^*\|^2 - 2\gamma \langle \mathcal{F}(\hat{\mathbf{x}}, \hat{\mathbf{y}}), (\hat{\mathbf{x}}, \hat{\mathbf{y}}) - \mathbf{z}^* \rangle \leq \|\bar{\mathbf{z}} - \mathbf{z}^*\|^2 . \qquad (8)$$

Finally, in view of

$$\langle \mathcal{F}(\hat{\mathbf{x}}, \hat{\mathbf{y}}), (\hat{\mathbf{x}}, \hat{\mathbf{y}}) - (\widetilde{\mathbf{x}}, \widetilde{\mathbf{y}}) \rangle$$

$$\overset{(7)}{\leq} \frac{1}{2\gamma} \|\bar{\mathbf{z}} - (\widetilde{\mathbf{x}}, \widetilde{\mathbf{y}})\|^2 - \frac{1}{2\gamma} \|\mathbf{z}_+ - (\widetilde{\mathbf{x}}, \widetilde{\mathbf{y}})\|^2$$

$$= \frac{1}{2\gamma} \left( \|\bar{\mathbf{x}} - \widetilde{\mathbf{x}}\|^2 - \|\mathbf{x}_+ - \widetilde{\mathbf{x}}\|^2 + \|\bar{\mathbf{y}} - \widetilde{\mathbf{y}}\|^2 - \|\mathbf{y}_+ - \widetilde{\mathbf{y}}\|^2 \right)$$

$$\leq \frac{1}{2\gamma} \left( \|\bar{\mathbf{x}} - \widetilde{\mathbf{x}} + \mathbf{x}_+ - \widetilde{\mathbf{x}}\| \cdot \|\bar{\mathbf{x}} - \mathbf{x}_+\| + \|\bar{\mathbf{y}} - \widetilde{\mathbf{y}} + \mathbf{y}_+ - \widetilde{\mathbf{y}}\| \cdot \|\bar{\mathbf{y}} - \mathbf{y}_+\| \right)$$

$$\leq \frac{1}{\gamma} \left( D_X \|\bar{\mathbf{x}} - \mathbf{x}_+\| + D_Y \|\bar{\mathbf{y}} - \mathbf{y}_+\| \right) \qquad (9)$$

$$\leq \frac{1}{\gamma} \sqrt{D_X^2 + D_Y^2} \cdot \sqrt{\|\bar{\mathbf{x}} - \mathbf{x}_+\|^2 + \|\bar{\mathbf{y}} - \mathbf{y}_+\|^2}$$

$$\leq \frac{1}{\gamma} \sqrt{D_X^2 + D_Y^2} \cdot \sqrt{2 \|\bar{\mathbf{z}} - \mathbf{z}^*\|^2 + 2 \|\mathbf{z}_+ - \mathbf{z}^*\|^2}$$

$$\overset{(8)}{\leq} \frac{2}{\gamma} \sqrt{D_X^2 + D_Y^2} \cdot \|\bar{\mathbf{z}} - \mathbf{z}^*\| ,$$

we have

$$u_1(\widetilde{\mathbf{x}}, \hat{\mathbf{x}}) - u_1(\hat{\mathbf{x}}, \hat{\mathbf{y}}) + u_2(\hat{\mathbf{x}}, \widetilde{\mathbf{y}}) - u_2(\hat{\mathbf{x}}, \hat{\mathbf{y}}) \leq \langle \mathcal{F}(\hat{\mathbf{x}}, \hat{\mathbf{y}}), (\hat{\mathbf{x}}, \hat{\mathbf{y}}) - (\widetilde{\mathbf{x}}, \widetilde{\mathbf{y}}) \rangle$$

$$\overset{(9)}{\leq} \frac{2}{\gamma} \sqrt{D_X^2 + D_Y^2} \cdot \|\bar{\mathbf{z}} - \mathbf{z}^*\|,$$

and the desired bound follows because $\widetilde{\mathbf{x}}$ and $\widetilde{\mathbf{y}}$ can take arbitrary points in $X$ and $Y$, respectively. $\square$

We also state the following sufficient condition for the accurate Nash equilibrium, which can be used as stopping criterion for the optimization algorithms. Similar results can be found, for instance, in Nemirovski (2004); Yang et al. (2020).

**Proposition 11** (Stopping criterion Nemirovski (2004); Yang et al. (2020)). *In a monotone general-sum game, let $\mathbf{z}^* = (\mathbf{x}^*, \mathbf{y}^*)$ be the Nash equilibrium. Let $\bar{\mathbf{z}} = (\bar{\mathbf{x}}, \bar{\mathbf{y}}) \in X \times Y$, $\gamma \in (0, \frac{1}{2L}]$, and $\mu = \min\{\mu, \nu\}$. We have*

$$\|\bar{\mathbf{z}} - \mathbf{z}^*\|^2 \leq \left( \frac{4}{\mu^2 \gamma^2} - \frac{2}{\mu\gamma} + 16 \right) \|\mathbf{z}_+ - \bar{\mathbf{z}}\|^2,$$

*where $\mathbf{z}_+ = \Pi_Z (\bar{\mathbf{z}} - \gamma \mathcal{F}(\hat{\mathbf{z}}))$, in which $\hat{\mathbf{z}} = \Pi_Z (\bar{\mathbf{z}} - \gamma \mathcal{F}(\bar{\mathbf{z}}))$.*

*Proof.* We have

$$(1 - \frac{\mu\gamma}{2}) \|\bar{\mathbf{z}} - \mathbf{z}^*\|^2 - \left( \frac{2}{\mu\gamma} - 1 \right) \|\mathbf{z}_+ - \bar{\mathbf{z}}\|^2$$

$$\leq \|\mathbf{z}_+ - \mathbf{z}^*\|^2$$

$$\overset{(7)}{\leq} \|\bar{\mathbf{z}} - \mathbf{z}^*\|^2 - 2\gamma \langle \mathcal{F}(\hat{\mathbf{z}}), \hat{\mathbf{z}} - \mathbf{z}^* \rangle$$

$$\leq \|\bar{\mathbf{z}} - \mathbf{z}^*\|^2 - 2\gamma \langle \mathcal{F}(\hat{\mathbf{z}}) - \mathcal{F}(\mathbf{z}^*), \hat{\mathbf{z}} - \mathbf{z}^* \rangle$$

$$\leq \|\bar{\mathbf{z}} - \mathbf{z}^*\|^2 - 2\mu\gamma \|\hat{\mathbf{z}} - \mathbf{z}^*\|^2$$

$$\leq \|\bar{\mathbf{z}} - \mathbf{z}^*\|^2 - \mu\gamma \|\bar{\mathbf{z}} - \mathbf{z}^*\|^2 + 2\mu\gamma \|\hat{\mathbf{z}} - \bar{\mathbf{z}}\|^2$$

$$= (1 - \mu\gamma) \|\bar{\mathbf{z}} - \mathbf{z}^*\|^2 - 2\mu\gamma \|\hat{\mathbf{z}} - \bar{\mathbf{z}}\|^2 + 4\mu\gamma \|\hat{\mathbf{z}} - \bar{\mathbf{z}}\|^2$$

$$\leq (1 - \mu\gamma) \|\bar{\mathbf{z}} - \mathbf{z}^*\|^2 - 2\mu\gamma \|\hat{\mathbf{z}} - \bar{\mathbf{z}}\|^2 + 8\mu\gamma \|\mathbf{z}_+ - \bar{\mathbf{z}}\|^2 + 8\mu\gamma \|\mathbf{z}_+ - \hat{\mathbf{z}}\|^2$$

$$\leq (1 - \mu\gamma) \|\bar{\mathbf{z}} - \mathbf{z}^*\|^2 - 2\mu\gamma \|\hat{\mathbf{z}} - \bar{\mathbf{z}}\|^2 + 8\mu\gamma \|\mathbf{z}_+ - \bar{\mathbf{z}}\|^2 + 8\mu\gamma \|\bar{\mathbf{z}} - \gamma\mathcal{F}(\hat{\mathbf{z}}) - \bar{\mathbf{z}} - \gamma\mathcal{F}(\bar{\mathbf{z}})\|^2$$

$$\leq (1 - \mu\gamma) \|\bar{\mathbf{z}} - \mathbf{z}^*\|^2 - 2\mu\gamma \|\hat{\mathbf{z}} - \bar{\mathbf{z}}\|^2 + 8\mu\gamma \|\mathbf{z}_+ - \bar{\mathbf{z}}\|^2 + 8\mu L^2 \gamma^3 \|\hat{\mathbf{z}} - \bar{\mathbf{z}}\|^2$$

$$\leq (1 - \mu\gamma) \|\bar{\mathbf{z}} - \mathbf{z}^*\|^2 + 8\mu\gamma \|\mathbf{z}_+ - \bar{\mathbf{z}}\|^2,$$

where in the second to last inequality we use $\gamma \leq \frac{1}{2L}$. Finally, the desired bound follows from rearrangement. $\square$

## C  DISCUSSIONS ON THE CATALYST METHODS

In this section, we present the intuition of most existing algorithms for convex-concave minimax optimization considering general conditioning, and explain why similar idea may not work directly when generalized to monotone near-zero-sum games.

Most of existing algorithms for minimax optimization with general conditioning are based on Catalyst (Lin et al., 2018). In minimax optimization, we have $u_1 + u_2 = 0$. Assume without loss of generality that $\mu \leq \nu$. The function $f(\mathbf{x}) \overset{\text{def}}{=} -u_1(\mathbf{x}, \mathbf{y}(\mathbf{x}))$ is $\mu$-strongly convex over $\mathbf{x} \in X$, in which $\mathbf{y}(\mathbf{x}) \overset{\text{def}}{=} \arg\max_{\mathbf{y} \in Y} u_2(\mathbf{x}, \mathbf{y})$. At the core of these algorithms, they build a function $\hat{f}_t$ and get an inexact solution $\hat{\mathbf{x}}_{t+1}$ at each iteration $t$:

$$\hat{\mathbf{x}}_{t+1} \approx \arg\min_{\mathbf{x} \in X} \left[ \hat{f}_t(\mathbf{x}) \overset{\text{def}}{=} f(\mathbf{x}) + \frac{\nu}{2} \|\mathbf{x} - \hat{\mathbf{x}}_t\|^2 \right]. \tag{10}$$

The outer loop is an inexact accelerated proximal point algorithm with $\widetilde{\mathcal{O}}\left(\sqrt{\frac{\nu}{\mu}} \cdot \log\left(\frac{1}{\varepsilon}\right)\right)$ iterations (Nesterov, 2005; Lin et al., 2018; Carmon et al., 2022), and the inner loop of solving the smoothed Equation (10) can be any method with the number of gradient queries $\widetilde{\mathcal{O}}\left(\frac{L}{\nu} \cdot \log\left(\frac{1}{\varepsilon}\right)\right)$ (Tseng, 1995). So, the total gradient complexity is[5]

$$\widetilde{\mathcal{O}}\left(\sqrt{\frac{\nu}{\mu}} \cdot \log\left(\frac{1}{\varepsilon}\right)\right) \cdot \widetilde{\mathcal{O}}\left(\frac{L}{\nu} \cdot \log\left(\frac{1}{\varepsilon}\right)\right) = \widetilde{\mathcal{O}}\left(\frac{L}{\sqrt{\mu\nu}} \cdot \log^2\left(\frac{1}{\varepsilon}\right)\right).$$

However, if we try to apply the above Catalyst method to monotone non-zero-sum games, the algorithm may only converge to a Stackelberg solution, which can be very different from the Nash equilibrium in non-zero-sum games.

EXAMPLE 3 (Stackelberg solution). *Consider the case where* $X = [0,1] \times [1,2] \subseteq \mathbb{R}^2$ *and* $Y = [-1,0] \subseteq \mathbb{R}$. *Let Player 1 maximize*

$$u_1(\mathbf{x}, y) = -\frac{1}{2}(x_1 - 1)^2 - \frac{1}{2}(x_2 - 1)^2 + \frac{1}{2}x_1 y$$

*over* $\mathbf{x} \in X$, *and Player 2 maximize*

$$u_2(\mathbf{x}, y) = \frac{1}{2}x_2 y - (y+1)^2$$

*over* $y \in Y$. *Then, the* Catalyst *minimization of* $f(x) = -u_1(\mathbf{x}, y(\mathbf{x}))$ *will lead to the Stackelberg solution* $(\mathbf{x} = \left(\frac{40}{63}, \frac{68}{63}\right), y = -\frac{46}{63})$, *which is different from the Nash equilibrium* $(\mathbf{x} = \left(\frac{5}{8}, 1\right), y = -\frac{3}{4})$.

Therefore, we are not aware of how the Catalyst methods for convex-concave minimax optimization can be applied in NEPs for non-zero-sum games.

## D    PROOF DETAILS

### D.1    PROOFS FOR THE RESULTS IN SECTION 3.1

*Proof of Proposition 3.* For any $\mathbf{z} = (\mathbf{x}, \mathbf{y}) \in X \times Y$,

$$\Delta(\mathbf{z}) \geq g(\mathbf{z}) - g(\mathbf{z}) + h(\mathbf{x}, \mathbf{y}) - h(\mathbf{x}, \mathbf{y}) = 0,$$

and for all $\widetilde{\mathbf{z}} = (\widetilde{\mathbf{x}}, \widetilde{\mathbf{y}}) \in X \times Y$, we have

$$\begin{aligned}
\Delta(\mathbf{z}) &\geq \frac{1}{2}\left[g(\mathbf{z}) - g(\mathbf{x}, \widetilde{\mathbf{y}}) + h(\mathbf{x}, \widetilde{\mathbf{y}}) - h(\mathbf{x}, \mathbf{y})\right] + \frac{1}{2}\left[g(\mathbf{z}) - g(\widetilde{\mathbf{x}}, \mathbf{y}) + h(\mathbf{x}, \mathbf{y}) - h(\widetilde{\mathbf{x}}, \mathbf{y})\right] \\
&= \frac{1}{2}\left[2g(\mathbf{z}) + u_2(\mathbf{x}, \widetilde{\mathbf{y}}) + u_1(\widetilde{\mathbf{x}}, \mathbf{y})\right] \\
&= \frac{1}{2}\left[u_1(\widetilde{\mathbf{x}}, \mathbf{y}) - u_1(\mathbf{x}, \mathbf{y}) + u_2(\mathbf{x}, \widetilde{\mathbf{y}}) - u_2(\mathbf{x}, \mathbf{y})\right].
\end{aligned}$$

$\square$

*Proof of Proposition 4.* The (if) part follows directly from Proposition 3. Now we prove the (only if) part. Suppose $\mathbf{z}^* = (\mathbf{x}^*, \mathbf{y}^*)$ is the Nash equilibrium. For all $\widetilde{\mathbf{z}} = (\widetilde{\mathbf{x}}, \widetilde{\mathbf{y}}) \in X \times Y$,

$$g(\mathbf{z}^*) - g(\widetilde{\mathbf{z}}) + h(\mathbf{x}^*, \widetilde{\mathbf{y}}) - h(\widetilde{\mathbf{x}}, \mathbf{y}^*) \leq \langle \nabla g(\mathbf{z}^*), \mathbf{z}^* - \widetilde{\mathbf{z}} \rangle + \langle \mathcal{H}(\mathbf{z}^*), \mathbf{z}^* - \widetilde{\mathbf{z}} \rangle \leq 0,$$

where in the first inequality we use Assumptions 1 and 2. Then, we have $\Delta(\mathbf{z}^*) = 0$.    $\square$

---

[5]The double logarithm term may be avoided by combining this algorithmic idea with some complicated techniques (Kovalev & Gasnikov, 2022; Carmon et al., 2022), which we omit here for the simplicity of presentation.

### D.2 PROOFS FOR THE RESULTS IN SECTION 3.2

The main technical work in the convergence analysis is to use the properties of our potential function and prove the descent lemma (Lemma 5).

*Proof of Lemma 5.* By Assumption 1, we can upper bound the convex-concave zero-sum part

$$
\begin{aligned}
h(\mathbf{x}_{t+1}, \mathbf{y}^*) - h(\mathbf{x}^*, \mathbf{y}_{t+1}) &= h(\mathbf{x}_{t+1}, \mathbf{y}_{t+1}) - h(\mathbf{x}^*, \mathbf{y}_{t+1}) + h(\mathbf{x}_{t+1}, \mathbf{y}^*) - h(\mathbf{x}_{t+1}, \mathbf{y}_{t+1}) \\
&\leq \langle \nabla_{\mathbf{x}} h(\mathbf{x}_{t+1}, \mathbf{y}_{t+1}), \mathbf{x}_{t+1} - \mathbf{x}^* \rangle - \frac{\mu}{2} \|\mathbf{x}_{t+1} - \mathbf{x}^*\|^2 \\
&\quad - \langle \nabla_{\mathbf{y}} h(\mathbf{x}_{t+1}, \mathbf{y}_{t+1}), \mathbf{y}_{t+1} - \mathbf{y}^* \rangle - \frac{\nu}{2} \|\mathbf{y}_{t+1} - \mathbf{y}^*\|^2 \\
&= \langle \mathcal{H}(\mathbf{z}_{t+1}), \mathbf{z}_{t+1} - \mathbf{z}^* \rangle - \frac{\mu}{2} \|\mathbf{x}_{t+1} - \mathbf{x}^*\|^2 - \frac{\nu}{2} \|\mathbf{y}_{t+1} - \mathbf{y}^*\|^2 .
\end{aligned}
\tag{11}
$$

By Assumptions 2 and 3, we can upper bound the jointly convex coupling part

$$
\begin{aligned}
g(\mathbf{z}_{t+1}) - g(\mathbf{z}^*) &= g(\mathbf{z}_{t+1}) - g(\mathbf{z}_t) + g(\mathbf{z}_t) - g(\mathbf{z}^*) \\
&\leq \langle \nabla g(\mathbf{z}_t), \mathbf{z}_{t+1} - \mathbf{z}_t \rangle + \frac{\delta}{2} \|\mathbf{z}_{t+1} - \mathbf{z}_t\|^2 + \langle \nabla g(\mathbf{z}_t), \mathbf{z}_t - \mathbf{z}^* \rangle \\
&= \langle \nabla g(\mathbf{z}_t), \mathbf{z}_{t+1} - \mathbf{z}^* \rangle + \frac{\delta}{2} \|\mathbf{z}_{t+1} - \mathbf{z}_t\|^2 .
\end{aligned}
\tag{12}
$$

In view of

$$
\langle \mathbf{z}_{t+1} - \mathbf{z}_t, \mathbf{z}_{t+1} - \mathbf{z}^* \rangle = \frac{1}{2} \|\mathbf{z}_{t+1} - \mathbf{z}^*\|^2 - \frac{1}{2} \|\mathbf{z}_t - \mathbf{z}^*\|^2 + \frac{1}{2} \|\mathbf{z}_{t+1} - \mathbf{z}_t\|^2 ,
\tag{13}
$$

and

$$
\begin{aligned}
&\left\langle \nabla g(\mathbf{z}_t) + \mathcal{H}(\mathbf{z}_{t+1}) + \frac{1}{\eta_t}(\mathbf{z}_{t+1} - \mathbf{z}_t), \mathbf{z}_{t+1} - \mathbf{z}^* \right\rangle \\
&\leq \langle \nabla_{\mathbf{x}} \varphi_t(\mathbf{z}_{t+1}), \mathbf{x}_{t+1} - \mathbf{x}^* \rangle - \langle \nabla_{\mathbf{y}} \varphi_t(\mathbf{z}_{t+1}), \mathbf{y}_{t+1} - \mathbf{y}^* \rangle \\
&\overset{(2)}{\leq} \varepsilon_t ,
\end{aligned}
\tag{14}
$$

we have

$$
\begin{aligned}
0 = -\Delta(\mathbf{z}^*) &\leq g(\mathbf{z}_{t+1}) - g(\mathbf{z}^*) + h(\mathbf{x}_{t+1}, \mathbf{y}^*) - h(\mathbf{x}^*, \mathbf{y}_{t+1}) \\
&\overset{(11)(12)}{\leq} \langle \nabla g(\mathbf{z}_t) + \mathcal{H}(\mathbf{z}_{t+1}), \mathbf{z}_{t+1} - \mathbf{z}^* \rangle - \frac{\mu}{2} \|\mathbf{x}_{t+1} - \mathbf{x}^*\|^2 - \frac{\nu}{2} \|\mathbf{y}_{t+1} - \mathbf{y}^*\|^2 + \frac{\delta}{2} \|\mathbf{z}_{t+1} - \mathbf{z}_t\|^2 \\
&= \left\langle \nabla g(\mathbf{z}_t) + \mathcal{H}(\mathbf{z}_{t+1}) + \frac{1}{\eta_t}(\mathbf{z}_{t+1} - \mathbf{z}_t), \mathbf{z}_{t+1} - \mathbf{z}^* \right\rangle - \frac{1}{\eta_t} \langle \mathbf{z}_{t+1} - \mathbf{z}_t, \mathbf{z}_{t+1} - \mathbf{z}^* \rangle \\
&\quad - \frac{\mu}{2} \|\mathbf{x}_{t+1} - \mathbf{x}^*\|^2 - \frac{\nu}{2} \|\mathbf{y}_{t+1} - \mathbf{y}^*\|^2 + \frac{\delta}{2} \|\mathbf{z}_{t+1} - \mathbf{z}_t\|^2 \\
&\overset{(13)(14)}{\leq} \varepsilon_t + \frac{1}{2\eta_t} \|\mathbf{x}_t - \mathbf{x}^*\|^2 - \left( \frac{1}{2\eta_t} + \frac{\mu}{2} \right) \|\mathbf{x}_{t+1} - \mathbf{x}^*\|^2 \\
&\quad + \frac{1}{2\eta_t} \|\mathbf{y}_t - \mathbf{y}^*\|^2 - \left( \frac{1}{2\eta_t} + \frac{\nu}{2} \right) \|\mathbf{y}_{t+1} - \mathbf{y}^*\|^2 - \left( \frac{1}{2\eta_t} - \frac{\delta}{2} \right) \|\mathbf{z}_{t+1} - \mathbf{z}_t\|^2 ,
\end{aligned}
$$

where the first equality follows from Proposition 4 and the first inequality follows from the definition of $\Delta(\cdot)$. Finally, the desired bound follows from $\eta_t \leq \frac{1}{\delta}$. $\qquad \square$

With Lemma 5, we are ready to prove the complexity of the outer loop (Lemma 6).

*Proof of Lemma 6.* For monotone $\delta$-nearly-zero-sum games and $\eta \leq \frac{1}{\delta}$, by Lemma 5, for any $k \in [0, t-1] \cap \mathbb{Z}$, we have

$$
\|\mathbf{z}_{k+1} - \mathbf{z}^*\|^2 \leq (1 - \theta) \|\mathbf{z}_k - \mathbf{z}^*\|^2 + 2\eta \varepsilon_k .
$$

Then, unrolling this recursion (from $k = t - 1, t - 2, \cdots$, to 0) yields

$$
\begin{aligned}
\|\mathbf{z}_t - \mathbf{z}^*\|^2 &\leq (1 - \theta)^t \|\mathbf{z}_0 - \mathbf{z}^*\|^2 + 2\eta \sum_{k=0}^{t-1} (1 - \theta)^{t-k-1} \varepsilon_k \\
&\leq (1 - \theta)^t (D_X^2 + D_Y^2) + \frac{2\eta}{\theta} \cdot \max_{k \in [0, t-1] \cap \mathbb{Z}} \varepsilon_k \\
&\leq \frac{\varepsilon}{2} + \frac{\varepsilon}{2} \\
&= \varepsilon \,,
\end{aligned}
$$

where the last inequality follows from $t \geq \frac{1}{\theta} \log \frac{2(D_X^2 + D_Y^2)}{\varepsilon}$ and $\varepsilon_t \leq \frac{\theta \varepsilon}{4\eta}$. $\qquad\square$

Below, we also include the proof of the gradient complexity of the inner loops for completeness. This result of the inner loops is heavily based on the previous results of optimal gradient methods in minimax optimization (see, for instance, Kovalev & Gasnikov (2022); Carmon et al. (2022); Thekumparampil et al. (2022); Lan & Li (2023)).

*Proof of Lemma 7.* Let $\mathbf{z}_{t+1}^* = (\mathbf{x}_{t+1}^*, \mathbf{y}_{t+1}^*) \in X \times Y$ denote the saddle point of $\varphi_t(\cdot, \cdot)$. Denote

$$
\bar{\varepsilon}_t = \frac{\varepsilon_t^2}{8L^2 (D_X^2 + D_Y^2)} \,.
$$

By Proposition 10, an inexact solution in Equation (2) of Algorithm 1 can be obtained from a pair of decisions $\bar{\mathbf{z}}_{t+1} = (\bar{\mathbf{x}}_{t+1}, \bar{\mathbf{y}}_{t+1}) \in X \times Y$ that satisfies $\left\| \bar{\mathbf{z}}_{t+1} - \mathbf{z}_{t+1}^* \right\|^2 \leq \bar{\varepsilon}_t$.

The function $\varphi_t(\cdot, \cdot)$ is $(\eta_t^{-1} + \mu)$-strongly convex-$(\eta_t^{-1} + \nu)$-strongly concave and $2L$-smooth, where the $2L$-smoothness follows from $\eta_t \geq \frac{1}{L}$. Hence, by Kovalev & Gasnikov (2022); Carmon et al. (2022); Thekumparampil et al. (2022); Lan & Li (2023), the aforementioned pair of decisions $\bar{\mathbf{z}}_{t+1}$ can be found within

$$
\mathcal{O}\left( \frac{L}{\sqrt{(\eta_t^{-1} + \mu)(\eta_t^{-1} + \nu)}} \cdot \log\left( \frac{D_X^2 + D_Y^2}{\bar{\varepsilon}_t} \right) \right)
$$

gradient queries. Finally, after substituting the $\bar{\varepsilon}_t$, the desired bound follows. $\qquad\square$

Finally, we prove Theorem 1, our main theoretical result.

*Proof of Theorem 1.* The overall gradient complexity is given by the multiplication of outer loop iterations (Lemma 6) and inner loop gradient complexity (Lemma 7):

$$
\begin{aligned}
&\mathcal{O}\left( \frac{\eta^{-1} + \min\{\mu, \nu\}}{\min\{\mu, \nu\}} \cdot \log \frac{2(D_X^2 + D_Y^2)}{\varepsilon} \right) \cdot \mathcal{O}\left( \frac{L}{\sqrt{(\eta^{-1} + \mu)(\eta^{-1} + \nu)}} \cdot \log \frac{L(D_X^2 + D_Y^2)}{\varepsilon_t} \right) \\
&= \mathcal{O}\left( \frac{\delta + \min\{\mu, \nu\}}{\min\{\mu, \nu\}} \cdot \log \frac{D_X^2 + D_Y^2}{\varepsilon} \right) \cdot \mathcal{O}\left( \frac{L}{\sqrt{(\delta + \mu)(\delta + \nu)}} \cdot \log\left( \frac{L(D_X^2 + D_Y^2)}{\min\{\mu, \nu\} \cdot \varepsilon} \right) \right) \\
&= \mathcal{O}\left( \frac{L}{\min\{\mu, \nu\}} \cdot \sqrt{\frac{\delta + \min\{\mu, \nu\}}{\delta + \max\{\mu, \nu\}}} \cdot \log\left( \frac{L(D_X^2 + D_Y^2)}{\min\{\mu, \nu\} \cdot \varepsilon} \right) \log\left( \frac{D_X^2 + D_Y^2}{\varepsilon} \right) \right) \\
&= \mathcal{O}\left( \left( \frac{L}{\sqrt{\mu\nu}} + \frac{L}{\min\{\mu, \nu\}} \cdot \min\left\{ 1, \sqrt{\frac{\delta}{\mu + \nu}} \right\} \right) \cdot \log\left( \frac{L(D_X^2 + D_Y^2)}{\min\{\mu, \nu\} \cdot \varepsilon} \right) \log\left( \frac{D_X^2 + D_Y^2}{\varepsilon} \right) \right) \,,
\end{aligned}
$$

where the first relation follows from $\eta = \min\left\{ \frac{1}{\delta}, \frac{1}{\min\{\mu, \nu\}} \right\}$. $\qquad\square$

### D.3 PROOFS FOR THE RESULT IN SECTION 3.3

We state a more general formulation of Corollary 9.

**Corollary 12.** *For monotone $\delta$-near-zero-sum games where $\mu = 0$ or $\nu = 0$, an $\varepsilon$-approximate Nash equilibrium can be found within*

$$\mathcal{O}\left(\left(\frac{L}{\sqrt{\bar{\mu}\bar{\nu}}} + \frac{L}{\min\{\bar{\mu}, \bar{\nu}\}} \cdot \min\left\{1, \sqrt{\frac{\delta}{\bar{\mu} + \bar{\nu}}}\right\}\right) \cdot \log\left(\frac{L^2 D^2}{\min\{\bar{\mu}, \bar{\nu}\} \cdot \varepsilon}\right) \log\left(\frac{LD^2}{\varepsilon}\right)\right)$$

*gradient queries, where $\bar{\mu} = \mu + \min\left\{\frac{\varepsilon}{2D_X^2}, L\right\}$ and $\bar{\nu} = \nu + \min\left\{\frac{\varepsilon}{2D_Y^2}, L\right\}$.*

*Proof of Corollary 12.* We consider the reduced game where Player 1 maximizes

$$\hat{u}_1 = u_1 - \min\left\{\frac{\varepsilon}{4D_X^2}, \frac{L}{2}\right\} \|\mathbf{x}\|^2 + \min\left\{\frac{\varepsilon}{4D_Y^2}, \frac{L}{2}\right\} \|\mathbf{y}\|^2$$

over $\mathbf{x} \in X$ and Player 2 maximizes

$$\hat{u}_2 = u_2 + \min\left\{\frac{\varepsilon}{4D_X^2}, \frac{L}{2}\right\} \|\mathbf{x}\|^2 - \min\left\{\frac{\varepsilon}{4D_Y^2}, \frac{L}{2}\right\} \|\mathbf{y}\|^2$$

over $\mathbf{y} \in Y$. Any $\frac{\varepsilon}{2}$-approximate Nash equilibrium of the reduced game is an $\varepsilon$-approximate Nash equilibrium in the original game. This reduction is similar to the ones used in Lin et al. (2020); Wang & Li (2020).

Denote $\hat{g} \triangleq -\frac{1}{2}(\hat{u}_1 + \hat{u}_2)$ and $\hat{h} \triangleq \frac{1}{2}(-\hat{u}_1 + u_2)$. Then, we have $\hat{h} = h + \left\{\frac{\varepsilon}{4D_X^2}, \frac{L}{2}\right\} \|\mathbf{x}\|^2 - \left\{\frac{\varepsilon}{4D_Y^2}, \frac{L}{2}\right\} \|\mathbf{y}\|^2$, which is $2L$-smooth and $\bar{\mu}$-strongly convex-$\bar{\nu}$-strongly concave. We also have $\hat{g} = -\frac{1}{2}(\hat{u}_1 + \hat{u}_2) = -\frac{1}{2}(u_1 + u_2) = g$, which is jointly convex and $\delta$-smooth. By Theorem 1, we obtain the number of gradient queries for an $\frac{\varepsilon^2}{32L^2D^2}$-accurate Nash equilibrium in the reduced game:

$$\mathcal{O}\left(\left(\frac{L}{\sqrt{\bar{\mu}\bar{\nu}}} + \frac{L}{\min\{\bar{\mu}, \bar{\nu}\}} \cdot \min\left\{1, \sqrt{\frac{\delta}{\bar{\mu} + \bar{\nu}}}\right\}\right) \cdot \log\left(\frac{L^2 D^2}{\min\{\bar{\mu}, \bar{\nu}\} \cdot \varepsilon}\right) \log\left(\frac{LD^2}{\varepsilon}\right)\right).$$

Finally, following from Proposition 10, we obtain the desired $\frac{\varepsilon}{2}$-approximate Nash equilibrium of the reduced game by taking an extragradient step from the $\frac{\varepsilon^2}{32L^2D^2}$-accurate Nash equilibrium. $\square$

### D.4 PROOFS FOR THE RESULTS IN SECTION 4

**Proposition 13** (Convex reformulation in bilinear coupling). *For $\beta_1, \beta_2 \geq 0$ and $\mathbf{M} \in R^{m \times n}$ such that $\sqrt{\beta_1 \beta_2} \geq \|\mathbf{M}\|$, the function $\widetilde{g}(\cdot, \cdot) : \mathbb{R}^n \times \mathbb{R}^m \to \mathbb{R}$ defined as*

$$\widetilde{g}(\mathbf{x}, \mathbf{y}) = \frac{\beta_1}{2} \|\mathbf{x}\|^2 + \langle \mathbf{M}\mathbf{x}, \mathbf{y} \rangle + \frac{\beta_2}{2} \|\mathbf{y}\|^2$$

*is jointly convex.*

*Proof.* The quadratic function $\widetilde{g}(\cdot, \cdot)$ is bounded below: for all $\mathbf{x} \in \mathbb{R}^n$, $\mathbf{y} \in \mathbb{R}^m$,

$$\widetilde{g}(\mathbf{x}, \mathbf{y}) \geq \frac{\beta_1}{2} \|\mathbf{x}\|^2 - \|\mathbf{M}\mathbf{x}\| \|\mathbf{y}\| + \frac{\beta_2}{2} \|\mathbf{y}\|^2$$

$$\geq \frac{\beta_1}{2} \|\mathbf{x}\|^2 - \sqrt{\beta_1 \beta_2} \|\mathbf{x}\| \|\mathbf{y}\| + \frac{\beta_2}{2} \|\mathbf{y}\|^2$$

$$\geq 0,$$

where in the first inequality we used the Cauchy-Schwarz inequality. Therefore, $\widetilde{g}(\cdot, \cdot)$ is jointly convex. $\square$

**Proposition 14** (Convex reformulation in general coupling). *For $\beta \geq 0$ and $g : X \times Y \to \mathbb{R}$ such that $g(\cdot, \cdot)$ is $\beta$-smooth, the function $\widetilde{g}(\cdot, \cdot) : X \times Y \to \mathbb{R}$ defined as*

$$\widetilde{g}(\mathbf{x}, \mathbf{y}) = \frac{\beta}{2} \|\mathbf{x}\|^2 + g(\mathbf{x}, \mathbf{y}) + \frac{\beta}{2} \|\mathbf{y}\|^2$$

*is jointly convex.*

*Proof.* For all $\mathbf{z} = (\mathbf{x}, \mathbf{y}) \in X \times Y$ and $\mathbf{z}' = (\mathbf{x}', \mathbf{y}') \in X \times Y$, we have

$$\begin{aligned}
\langle \nabla \widetilde{g}(\mathbf{z}') - \nabla \widetilde{g}(\mathbf{z}), \mathbf{z}' - \mathbf{z} \rangle &= \beta \|\mathbf{z}' - \mathbf{z}\|^2 + \langle \nabla g(\mathbf{z}') - \nabla g(\mathbf{z}), \mathbf{z}' - \mathbf{z} \rangle \\
&\geq \beta \|\mathbf{z}' - \mathbf{z}\|^2 - \beta \|\mathbf{z}' - \mathbf{z}\|^2 \\
&= 0 \,,
\end{aligned}$$

where the first inequality follows from the $\beta$-smoothness of $g(\cdot, \cdot)$. Therefore, the function $\widetilde{g}(\cdot, \cdot)$ is jointly convex (Nesterov, 2004, Theorem 2.1.3). $\qquad\square$

## E  ADDITIONAL RESULTS FOR OTHER ORACLE AND FUNCTION CLASSES

In this section, we consider a different class of Nash equilibrium problem and demonstrate the applicability of our ICL framework. In Boţ et al. (2023), they considered a zero-sum (or strictly competitive) game where the two players have proximal oracle and gradient oracle, respectively. We now consider the generalization where an additional incentive is added. To prevent ambiguity, we will define the problem class in a self-contained way.

In this section, we are interested in the Nash equilibrium problem, or equivalently, the variational inequality problem given by operator $\mathcal{F}$ defined on $X \times Y$:

$$\mathcal{F}(\mathbf{x}, \mathbf{y}) = \nabla g(\mathbf{x}, \mathbf{y}) + (\nabla_{\mathbf{x}} h(\mathbf{x}, \mathbf{y}), -\nabla_{\mathbf{y}} h(\mathbf{x}, \mathbf{y}) + \nabla \psi(\mathbf{y})) \,, \quad (\mathbf{x}, \mathbf{y}) \in X \times Y,$$

where

1. The sets $X$ and $Y$ are compact convex sets in Euclidean spaces. The diameter of $X$ is bounded by $D_X$, and the diameter of $Y$ is bounded by $D_Y$.
2. The function $g \colon X \times Y \to \mathbb{R}$ is $\delta$-smooth and convex.
3. The function $\psi \colon Y \to \mathbb{R} \cup \{+\infty\}$ is proper, lower semicontinuous, $\nu$-strongly convex, and with domain $\mathrm{dom}\, \psi = \{\mathbf{y} \in Y \mid \psi(y) < +\infty\}$.
4. For all $\mathbf{y} \in \mathrm{dom}\, \psi$, the function $h(\cdot, \mathbf{y}) \colon X \to \mathbb{R} \cup \{+\infty\}$ is proper, lower semi-continuous, and $\mu$-strongly convex.
5. For all $\mathbf{x} \in \Pi_X(\mathrm{dom}\, h) \triangleq \{\mathbf{u} \in X \mid \exists \mathbf{v} \in Y \text{ such that } (\mathbf{u}, \mathbf{v}) \in \mathrm{dom}\, h\}$, we have that $\mathrm{dom}\, h(\mathbf{x}, \cdot) = Y$ and the function $h(\mathbf{x}, \cdot) \colon Y \to \mathbb{R}$ is concave and continuously differentiable. Moreover, $\Pi_X(\mathrm{dom}\, h)$ is closed.
6. There exists $L_{yx}, L_{yy} \geq 0$ such that for all $(x, y), (x', y') \in \Pi_X(\mathrm{dom}\, h) \times \mathrm{dom}\, \psi$,

$$\|\nabla_{\mathbf{y}} h(\mathbf{x}, \mathbf{y}) - \nabla_{\mathbf{y}} h(\mathbf{x}', \mathbf{y}')\| \leq L_{yx} \|\mathbf{x} - \mathbf{x}'\| + L_{yy} \|\mathbf{y} - \mathbf{y}'\| \,.$$

We assume the players can query the gradient $\nabla g$, the proximal operator of $h(\cdot, \mathbf{y})$ for any fixed $\mathbf{y} \in Y$, the partial gradient $\nabla_{\mathbf{y}} h(\cdot, \cdot)$, and the proximal oracle of $\psi(\cdot)$. The problem studied in Boţ et al. (2023) corresponds to a special case of the additional incentive $\delta = 0$, while we generalize their results to $\delta \neq 0$.

We will use the complexity results in Boţ et al. (2023) as a black box. We cite their results below.

**Lemma 15** (Boţ et al. (2023), Theorem 14). *For $\delta = 0$ and $\mu > 0$, there exists an algorithm which returns an $\varepsilon$-accurate Nash equilibrium with the number of partial gradient queries to $\nabla_{\mathbf{y}} h(\cdot)$ and the number of proximal oracle queries to $\mathrm{prox}_{h(\cdot, \mathbf{y})}(\cdot)$ and $\mathrm{prox}_{\psi}(\cdot)$ bounded by*

$$\mathcal{O}\left( \left(1 + \frac{L_{yx}}{\sqrt{\mu\nu}} + \frac{L_{yy}}{\nu}\right) \cdot \log\left(\frac{D^2}{\varepsilon}\right) \right) \,.$$

By applying our ICL algorithm (Algorithm 1), we obtain the following complexity result:

**Theorem 2.** *Assume $h$ is $L$-smooth over $\Pi_X (\mathrm{dom}\, h) \times \mathrm{dom}\, \psi$, and $\psi$ is $L$-smooth over $\mathrm{dom}\, \psi$. For $\mu > 0$, there exists an algorithm which returns an $\varepsilon$-accurate Nash equilibrium with the number of partial gradient queries to $\nabla_{\mathbf{y}} h(\cdot)$ and the number of proximal oracle queries to $\mathrm{prox}_{h(\cdot, \mathbf{y})}(\cdot)$ and $\mathrm{prox}_{\psi}(\cdot)$ bounded by*

$$\mathcal{O}\left(\left(1 + \frac{\delta}{\min\{\mu, \nu\}}\right)\left(1 + \frac{L_{yx}}{\sqrt{(\delta + \mu)(\delta + \nu)}} + \frac{L_{yy}}{\delta + \nu}\right) \cdot \log\left(\frac{LD^2}{\min\{\mu, \nu\} \cdot \varepsilon}\right) \log\left(\frac{D^2}{\varepsilon}\right)\right),$$

*and with the number of gradient queries to $\nabla g(\cdot)$ bounded by*

$$\mathcal{O}\left(\left(1 + \frac{\delta}{\min\{\mu, \nu\}}\right) \cdot \log\left(\frac{D^2}{\varepsilon}\right)\right).$$

*Proof.* The result follows from multiplying the outer loop iterations in Lemma 6 and the inner loop complexity in Lemma 15. $\qquad\square$

## F  ILLUSTRATION OF THE APPLICATION EXAMPLES

**Matrix games with transaction fees**  We give a concrete illustration for matrix games with transaction fees. Let the payoff matrices of Player 1 and Player 2 without transaction fees be

$$\mathbf{M} = \begin{bmatrix} 300 & -200 \\ -100 & 400 \end{bmatrix} \quad \text{and} \quad -\mathbf{M} = \begin{bmatrix} -300 & 200 \\ 100 & -400 \end{bmatrix},$$

respectively. Then,

$$\mathrm{abs}\,(\mathbf{M}) = \begin{bmatrix} 300 & 200 \\ 100 & 400 \end{bmatrix}, \quad \mathbf{M}_+ = \begin{bmatrix} 300 & 0 \\ 0 & 400 \end{bmatrix}, \quad \text{and} \quad \mathbf{M}_- = \begin{bmatrix} 0 & 200 \\ 100 & 0 \end{bmatrix}.$$

Let $1\%$ of transaction fees be imposed on every payment. Then, the payoff matrices of Player 1 and Player 2 with transaction fees are

$$\mathbf{A} = \begin{bmatrix} 297 & -200 \\ -100 & 396 \end{bmatrix} \quad \text{and} \quad \mathbf{B} = \begin{bmatrix} -300 & 198 \\ 99 & -400 \end{bmatrix},$$

respectively. We also draw the Table 2 for easier comparisons.

Table 2: An illustration of matrix games with transaction fee $\rho = 0.01$.

| 300/-300 | -200/200 | | 297/-300 | -200/198 |
|---|---|---|---|---|
| -100/100 | 400/-400 | | -100/99 | 396/-400 |

**Competitive games with small cooperation incentives**  We give a concrete illustration for competitive games with small cooperation incentives. The donation games are a canonical model in evolutionary game theory for studying the altruistic collaboration (Nowak, 2006a; Sigmund, 2010).

Let us consider a simplified version of donation games played for one round, as shown in Table 3. In this game, a player can choose to incur a personal cost, $c$, to provide a larger benefit, $b$, to another player. Consider a concrete example where the cost to donate is $c = 50$ and the benefit conferred is only slightly higher at $b = 51$.

In this scenario, if both players cooperate, they each pay a cost of $50$ and receive a benefit of $51$, resulting in a modest net payoff of $1$. However, the temptation to defect is substantial: by withholding their own donation while still receiving the benefit from the other player, a defector can achieve a payoff of $51$. Conversely, the cooperating player who is defected upon is left with a "sucker's payoff" of $-50$. This low benefit-to-cost ratio ($b/c \approx 1.02$) creates a very small incentive for cooperation.

Table 3: An illustration of donation games.

|  | Player 2: Cooperate | Player 2: Defect |
|---|---|---|
| Player 1: Cooperate | (1, 1) | (-50, 51) |
| Player 1: Defect | (51, -50) | (0, 0) |

## G MORE EXPERIMENTAL DETAILS

### G.1 IMPLEMENTATION DETAILS

We generate the sparse matrix $\mathbf{M}$ following the procedures outlined in Nemirovski (2004); Nesterov (2005): (i) The random seeds are set from 0, 111, 222, ..., and 999; (ii) 100000 coordinates of $\mathbf{M}$ are chosen uniformly at random; (iii) each chosen coordinate is assigned a random value independently drawn from a uniform distribution between $[-1, 1]$; (iv) all remaining coordinates are set to 0.

We implement our ICL method as described in Algorithm 1. The classic OGDA and classic EG methods are implemented as outlined in Popov (1980) and Korpelevich (1976), respectively. All solvers are initialized at $(\mathbf{x}_0, \mathbf{y}_0) = (\mathbf{1}_n/n, \mathbf{1}_m/m)$, where $\mathbf{1}_k \in \mathbb{R}^k$ denotes the vector of size $k$ where every element in the vector is equal to 1. The setup for ICL is detailed in Theorem 1. The stepsize for OGDA is set to $\frac{1}{2L}$ following Popov (1980); Mokhtari et al. (2020), and for EG is set to $\frac{1}{\sqrt{2}L}$ following Korpelevich (1976); Nemirovski (2004). For the inner loop, the Lifted Primal Dual method (Thekumparampil et al., 2022) is used, with the theoretical setup maintained as specified in (Thekumparampil et al., 2022, Theorem 2).

### G.2 MORE DETAILS OF THE EXPERIMENT RUNS

We conducted our experiments on `e2-highcpu` vCPUs within the `Google Cloud` environment. The memory requirement of our experiments is quite modest, requiring only sufficient RAM for a few $10000 \times 10000$ sparse matrices (that is, about 60 MB). Each independent run completes within about 3 minutes.

We plot the convergence behaviors in Figure 1. Note that for ICL, only iterates within the outer loop are plotted. Figure 1 shows results for a single seed (seed 0), as plotting all seeds in a single figure would introduce excessive visual complexity due to the unaligned $x$-axis representing the counts of gradient queries in the outer loop. Nonetheless, we observed consistent convergence patterns across different seeds: (i) Transaction fee changes have little impact on the convergence of OGDA and EG, but significantly accelerate the convergence of ICL as $\rho$ decreases; (ii) ICL converges fastest when $\rho \leq 0.12\%$; and (iii) OGDA converges fastest when $\rho \geq 0.15\%$.

We should clarify that all 21 entries (7 entries for each of the three algorithms) are indeed plotted in Figure 1. The perceived "lack of distinct lines" is actually a crucial empirical result supporting our theoretical claims:

- For OGDA (Red) and EG (Green): The seven lines corresponding to different parameter settings for OGDA and EG almost completely merge and overlap. This empirical observation is fully consistent with our theory, which states that the convergence rate of classical variational inequality methods does not benefit from the near-zero-sum structure. Their rates are determined solely by $\frac{L}{\min\{\mu,\nu\}}$, regardless of the $\delta$ parameter.

- For the proposed ICL algorithm (Blue): The seven lines are clearly separated, showing a distinct dependency on the $\delta$ parameter. This separation empirically validates our key claim that our algorithm successfully harnesses the near-zero-sum structure to achieve a faster convergence rate.

Finally, we report CPU times of experiment runs to converge to an $\varepsilon$-accurate Nash equilibrium in Table 4, with error bars indicating 2-sigma variations across 10 independent runs using randomly generated matrices. Table 4 shows that ICL achieves the shortest CPU time when $\rho \leq 0.12\%$, while OGDA achieves the shortest CPU time when $\rho \geq 0.15\%$.

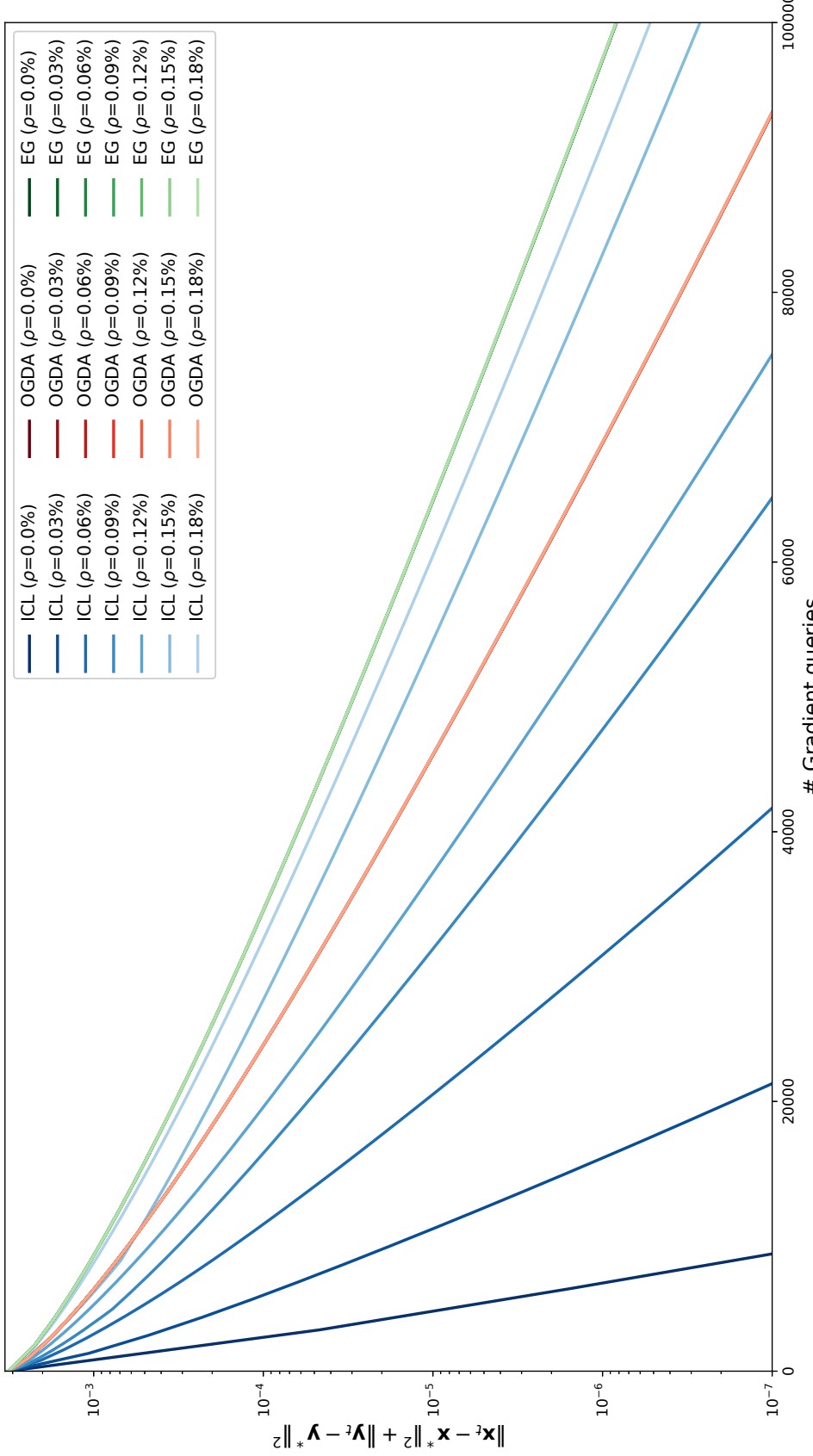

Figure 1: Comparisons of the convergence of the ICL, OGDA, and EG methods with respect to the gradient query counts.

Table 4: The CPU times (in seconds) of the algorithms to converge to an $\varepsilon$-accurate Nash equilibrium under various transaction fees. The error bars indicate 2-sigma variations across the independent runs with 10 randomly generated matrices.

| Transaction fee $\rho$ / Methods | 0.00% | 0.03% | 0.06% | 0.09% | 0.12% | 0.15% | 0.18% |
|---|---|---|---|---|---|---|---|
| ICL (Algorithm 1) | $\mathbf{19 \pm 0}$ | $\mathbf{49 \pm 0}$ | $\mathbf{93 \pm 0}$ | $\mathbf{142 \pm 1}$ | $\mathbf{167 \pm 0}$ | $247 \pm 2$ | $264 \pm 3$ |
| OGDA (Popov, 1980) | $186 \pm 1$ | $185 \pm 1$ | $185 \pm 1$ | $185 \pm 0$ | $185 \pm 1$ | $\mathbf{185 \pm 1}$ | $\mathbf{186 \pm 1}$ |
| EG (Korpelevich, 1976) | $258 \pm 1$ | $256 \pm 2$ | $258 \pm 2$ | $257 \pm 3$ | $257 \pm 2$ | $257 \pm 2$ | $257 \pm 2$ |

## G.3 ADDITIONAL RUNS UNDER DIFFERENT PARAMETER SETTING

In this section, we run additional numerical experiments under different parameter setting. We change the parameter $\nu = 0.01$, and we vary the transaction fee $\delta$ from $\{0.0\%, 0.3\%, \cdots, 1.8\%\}$. We keep the other parameter settings unchanged.

The results, summarized in Table 5, demonstrate that ICL requires fewer gradient queries to converge to an $\varepsilon$-accurate Nash equilibrium when the transaction fee $\rho$ is below $1.2\%$. This empirical observation aligns with our theoretical prediction in Example 1, which suggests that ICL converges faster when $\rho \left\| \mathrm{abs}\left(\mathbf{M}\right) \right\| \ll \sqrt{\mu\nu} = 10\%$.

Table 5: Gradient query counts to converge to an $\varepsilon$-accurate Nash equilibrium under various transaction fees. Error bars indicate 2-sigma variations across the independent runs with 10 randomly generated matrices.

| Transaction fee $\rho$ / Methods | 0.0% | 0.3% | 0.6% | 0.9% | 1.2% | 1.5% | 1.8% |
|---|---|---|---|---|---|---|---|
| ICL (Algorithm 1) | $\mathbf{924 \pm 0}$ | $\mathbf{924 \pm 0}$ | $\mathbf{824 \pm 0}$ | $\mathbf{1030 \pm 0}$ | $\mathbf{1236 \pm 0}$ | $1648 \pm 0$ | $2060 \pm 0$ |
| OGDA (Popov, 1980) | $1364 \pm 0$ | $1364 \pm 0$ | $1364 \pm 0$ | $1361 \pm 4$ | $1359 \pm 5$ | $\mathbf{1353 \pm 6}$ | $\mathbf{1350 \pm 6}$ |
| EG (Korpelevich, 1976) | $1848 \pm 0$ | $1848 \pm 0$ | $1848 \pm 0$ | $1848 \pm 0$ | $1848 \pm 0$ | $1848 \pm 0$ | $1848 \pm 0$ |

We also observe that the CPU times in this setting are within 5 seconds for all independent runs.

