# OpenReview forum: "Monotone Near-Zero-Sum Games: A Generalization of Convex-Concave Minimax"
_ICLR.cc/2026/Conference — ICLR 2026 Poster_

### Official Review · Reviewer_BPVS · 2025-10-24

**Soundness:** 3
**Presentation:** 4
**Contribution:** 3
**Rating:** 8
**Confidence:** 3

**Summary:**

This paper studies the problem of computing NE in monotone games beyond the zero-sum regime. In particular, the authors define the class of monotone $\delta$-near-zero-sum games, where the $\delta$ parameter denotes that the coupled component of the game $g(\cdot, \cdot)$ is $\delta$-smooth. This class interpolates between zero-sum ($\delta=0$) and general-sum ($\delta=L$) monotone games. While the zero-sum case has been well studied and admits minimax optimal algorithms, the general-case complexity from Tseng (1995) remains less well-studied. The paper describes a new algorithm called Iterative Coupling Linearization (ICL), which is shown to converge faster than existing methods when $\delta$ is small enough, in both strongly and non-strongly monotone games. The algorithm relies on a linearization of the coupling part $g$, leading to a sequence of zero-sum subproblems. Finally, the paper exhibits several applications including regularized matrix games and competitive games with small cooperation incentives. The experiments are run on randomized examples of the former class, showing corroboration for the theoretical result.

**Strengths:**

- The paper is very well written and clear. The research question studied is of clear interest to the community, and is well-motivated in the introduction.
- The use of the potential function and linearization in the design of ICL is a nice idea, and leads to improved/optimal convergence rates in certain game classes.
- The applications given are nicely motivated and meaningfully extend the class of games for which we can guarantee fast convergence to Nash equilibria. The subsequent study of other similarly computationally 'easy' games seems to be a fruitful direction.
- The technical components of the paper are clean and easy to digest.

**Weaknesses:**

- The presence of the quadratic log factor naturally brings up the question of whether this is an artifact of the algorithm proposed or a fundamental separation between zero-sum and non-zero-sum games. The paper contains only one 'main' theoretical result, which is itself not a weakness, but I would have liked to see more discussion or even some preliminary results towards establishing lower bounds. This is briefly mentioned in the concluding remarks, but I believe this should be given more focus by the authors. Can the authors comment on this?
- The experiments given only pertain to Example 1. The ICL algorithm would be more compelling if some results can be shown in other examples or game classes.

**Questions:**

- Is it correct to say that recognizing that a game is $\delta$-near-zero-sum is not a trivial task, since Assumption 3 is an existence statement? Are there other effective notions of closeness that are easier to verify?
- No-regret algorithms have seen some success in converging to NE in multiplayer zero-sum monotone games, can a similar result potentially be established for ICL? Specifically, ICL's design relies heavily on the minimax (i.e. two player) formulation -- is there an extension that can work for games with more players?

---

> ### Author Response · Authors · 2025-11-19
> **Response to Reviewer BPVS**
>
> We thank the reviewer for the positive feedback and for the interesting questions about the paper.
>
> > The presence of the quadratic log factor naturally brings up the question of whether this is an artifact of the algorithm proposed or a fundamental separation between zero-sum and non-zero-sum games.
>
> It is an insightful question whether the additional logarithm can be removed by a different algorithm or it is an intrinsic gap between zero-sum and non-zero-sum classes.
>
> We believe that the logarithm is introduced by the algorithm rather than by the problem class. In this work, we show in principle that the near-zero-sum games can be solved by the off-the-shelf zero-sum methods via our black-box reduction algorithm. Our algorithm comes with an additional logarithm term, which seems to be a reasonable price for this level of generality.
>
> On the more technical side, this additional logarithm resembles the additional log factors appearing in the family of proximal point methods. Removing the logarithm in proximal point methods often requires some slightly different stopping criterions (for instance, the Monteiro-Svaiter condition) for the inner loops. In our framework, removing this logarithm is an interesting question we leave for future work, as it requires opening up the black-box reduction to some extent to design more specific algorithms.
>
> > The paper contains only one 'main' theoretical result, which is itself not a weakness, but I would have liked to see more discussion or even some preliminary results towards establishing lower bounds. This is briefly mentioned in the concluding remarks, but I believe this should be given more focus by the authors. Can the authors comment on this?
>
> It is very interesting to have a lower bound for our new class.
> This is, however, a more difficult and very non-trivial problem. The matching lower bound for two-player zero-sum games is only established recently in [Zhang et al. 2022]. In fact, despite the long-standing upper complexity bounds from [Tseng 1995; Nemirovsky 2004], the lower bound for two-player general-sum classes with general conditioning still remains widely open.
>
> On the more technical side, the lower bound construction for zero-sum games [Zhang et al. 2022] is based on a quadratic function that resembles Nesterov’s “worst function in the world”. However, for our new class as well as for the general-sum class of $\nabla g + \mathcal H$, to show a tighter lower bound than zero-sum classes, we may need to create a *non-quadratic* bad function with general coupling. It is a difficult open problem and can be much more complicated than the quadratic constructions.
>
> In our revision, we added these discussions in the section of future work.
>
> > The experiments given only pertain to Example 1. The ICL algorithm would be more compelling if some results can be shown in other examples or game classes.
>
> **Please kindly check our general response on the basic numerical experiment.**
>
> > Is it correct to say that recognizing that a game is $\delta$-near-zero-sum is not a trivial task, since Assumption 3 is an existence statement? Are there other effective notions of closeness that are easier to verify?
>
> To avoid confusion, we changed Assumption 3 to “Let $\delta > 0$ such that the function $g$ is $\delta$ smooth.”
> **Please kindly check our general response on the estimation of $\delta$.**
>
> > No-regret algorithms have seen some success in converging to NE in multiplayer zero-sum monotone games, can a similar result potentially be established for ICL? Specifically, ICL's design relies heavily on the minimax (i.e. two player) formulation -- is there an extension that can work for games with more players?
>
> If we understand correctly, the reviewer was actually referring to the success in the multiplayer general-sum monotone games with equal conditioning. For the problem classes considered in our paper, this corresponds to the slow rates in [Tseng 1995; Nemirovsky 2004]).
> However, the fast rate of $L / \sqrt {\mu \nu}$ has never been extended to multiplayer games to our knowledge.
>
> **Please kindly check our general response on extensions to multiplayer games.** We give both conceptual and technical reasons why the multiplayer games are structurally different, and the extension can be highly non-trivial.
>
> We thank the reviewer for suggesting the discussion of multiplayer games, and we updated our section 6 to include these discussions.
>
> We hope the above addresses all the issues raised in the reviews and clarifies the interesting future direction of further extending monotone near-zero-sum games.

---

> > ### Comment · Reviewer_BPVS · 2025-11-25
> > **Response to rebuttal**
> >
> > Thank you for the detailed responses to the shared concerns of the reviewers. Thank you also for clarifying my confusion about the convergence rate of no-regret learning in monotone games considered in the work. While the difficulties of extending the results to multiplayer games and the dependence on $\delta$ lead to some practicality concerns, I am still happy with the theoretical/conceptual contribution of the work, and will maintain my score.

---

> > > ### Author Response · Authors · 2025-11-25
> > >
> > > Dear Reviewer BPVS: We are grateful for your suggestions on future extensions, many of which present interesting open challenges (even for the basic zero-sum class). Thank you again for the positive feedback and helpful discussion!

---

> ### Author Response · Authors · 2025-11-19
> **References**
>
> *References:*
>
> *Tseng, Paul. "On linear convergence of iterative methods for the variational inequality problem." Journal of Computational and Applied Mathematics 60.1-2 (1995): 237-252.*
>
> *Nemirovski, Arkadi. "Prox-method with rate of convergence O (1/t) for variational inequalities with Lipschitz continuous monotone operators and smooth convex-concave saddle point problems." SIAM Journal on Optimization 15.1 (2004): 229-251.*
>
> *Zhang, Junyu, Mingyi Hong, and Shuzhong Zhang. "On lower iteration complexity bounds for the convex concave saddle point problems." Mathematical Programming 194.1 (2022): 901-935.*

---

### Official Review · Reviewer_AbM8 · 2025-10-30

**Soundness:** 3
**Presentation:** 2
**Contribution:** 2
**Rating:** 6
**Confidence:** 3

**Summary:**

The paper introduces a new class of two player games, termed “Monotone Near-Zero-Sum Games”, in between the classes of zero sum and general sum games. The class consists of games where averaging the utilities of the two players results in a smooth function, with the smoothness parameter $\delta$ indicating the interpolation between the classes. One of the motivations of the paper for defining the class lies in the gap of gradient complexity, which is the theoretical focus of the paper, between zero and general sum games. The authors propose a novel algorithm whose complexity depends on $\delta$ and matches, up to logarithmic terms, the state of the art results for both classes. Finally, they present some experimentation comparing their algorithm with previous methods in some families of games.

**Strengths:**

On a conceptual level, the paper does take a step in filling the gap between zero sum and general sum games, introducing an interesting class. Moreover, from the technical side, the proofs require technical work. On the application side, the transaction fee games are interesting as a class so I appreciate a method that solves them efficiently.

**Weaknesses:**

I think that the weaker aspect of the paper lies in its applications. Aside from the example mentioned above, I am not sure about the practical relevance of the classes that can be captured within the presented framework.

**Questions:**

Could you provide some additional examples of games that can be solved with your approach? Or some more illustrations, especially for the competitive games with small additional incentives? Also, could you include some references about the transaction fee games, even outside the literature of gradient complexity, as well as comparisons with different approaches to solve them?

I will also add two minor comments:
Equation (13) is incorrect. It should be 2 instead of $1/2$. (The substitution later on is correct)
Figure 1 is somewhat confusing. Based on the text, I think that the best out of the 3 methods are depicted but that it is not clear from the figure itself. The legend is also confusing, with 21 entries but much fewer lines depicted. Also, it is not easy to tell some colors apart.

---

> ### Author Response · Authors · 2025-11-19
> **Response to Reviewer AbM8**
>
> We thank the reviewer for the helpful feedback and for the careful examination of the proof steps in our paper.
>
> > I think that the weaker aspect of the paper lies in its applications. Aside from the example mentioned above, I am not sure about the practical relevance of the classes that can be captured within the presented framework.
>
> > Could you provide some additional examples of games that can be solved with your approach? Or some more illustrations, especially for competitive games with small additional incentives? Also, could you include some references about the transaction fee games, even outside the literature of gradient complexity, as well as comparisons with different approaches to solve them?
>
> **Please kindly check our general response on the application examples.** We have also revised our paper in section 4 and Appendix F to provide detailed scenarios, more economic backgrounds, and additional citations.
>
> > Equation (13) is incorrect. It should be 2 instead of $\frac 1 2$. (The substitution later on is correct)
>
> We thank the reviewer for pointing out this typo in the analysis. We fixed it in the revision.
>
> > Figure 1 is somewhat confusing. Based on the text, I think that the best out of the 3 methods are depicted but that it is not clear from the figure itself. The legend is also confusing, with 21 entries but much fewer lines depicted. Also, it is not easy to tell some colors apart.
>
> We thank the reviewer for your nice observation on Figure 1. This is indeed a solid validation of our theory, and we have updated our paper to discuss this visual effect. **Please kindly check our general response on basic numerical experiments.**
>
> We hope our response answers the reviewer’s questions on the application examples and confusion on the numerical results.

---

### Official Review · Reviewer_F4W2 · 2025-10-31

**Soundness:** 4
**Presentation:** 3
**Contribution:** 2
**Rating:** 6
**Confidence:** 2

**Summary:**

In this paper, the authors targets on the gap between gradient complexity of monotone general game ($\mathcal{O}(\frac{L}{\min\{\mu, \nu\}}\cdot \log(D^2/\varepsilon)$) and monotone zero-sum game  ($\mathcal{O}(\frac{L}{\sqrt{\mu \nu}}\cdot \log(D^2/\varepsilon)$). To bridge this gap, the authors propose monotone near-zero-sum games , characterized by a smoothness paraemeter $\delta$ describing the game's proximity to a zero-sum game. Then, the author present Iterative Coupling Linearization (ICL) algorithm that provide black-box reduction from monotone near-zero sum game to  zero-sum games and derive gradient complexity in-between the known ones of zero-sum and general-sum games.

**Strengths:**

1. The problem is theoretically well-motivated. The authors clearly identify a gap between the gradient complexities of monotone zero-sum and monotone general-sum games, and take a principled step toward bridging this gap through the introduction of a new intermediate class of monotone near-zero-sum games.
2. The presentation is clear and well-structured. The decomposition of the potential function into a jointly convex coupling part and a convex–concave zero-sum part provides strong intuition that connects the problem formulation, algorithm design, and convergence analysis in a coherent way.
3. The proposed Iterative Coupling Linearization (ICL) algorithm is conceptually novel and algorithmically elegant, and the the convergence analysis is technically rigorous.

**Weaknesses:**

My main concern is generalizability of the method presented in the paper and the overall contribution of the paper.
1. The main theoretical results hinge on the near-zero-sum smoothness parameter $\delta$ that measures how strongly coupled the game is. However, it remains unclear in which practical settings the $\delta$-smoothness assumption naturally holds or how to estimate it empirically. Although it is formally bounded by the overall smoothness $L$, it is uncertain how often $\delta$ would be sufficiently small in realistic problems to yield the claimed acceleration in convergence.
2. In addition, the presented results are developed specifically for two-player games, whereas the classical gradient complexity results for monotone general-sum games (e.g., Tseng, 1995; Nemirovski, 2004) extend to arbitrary numbers of players. It would strengthen the contribution to discuss whether the proposed framework and analysis can be generalized to multi-player settings, or to explain the technical barriers that currently prevent such an extension.

**Questions:**

Would you expect a similar analysis and complexity improvement to extend to the multi-player setting? If not, could you elaborate on the main technical barriers that prevent such an extension? In particular, are there specific steps in the convergence proof or properties of the two-player potential function that fail to generalize when the number of players exceeds two?

Moreover, could you provide additional examples or intuitions for practical game settings where the coupling part $g$ is expected to have a small smoothness parameter $\delta$? In other words, in what types of real-world or commonly studied games would the near-zero-sum assumption $(\delta<<L)$ naturally hold?

---

> ### Author Response · Authors · 2025-11-19
> **Response to Reviewer F4W2**
>
> We thank the reviewer for the helpful feedback and for the suggestions on possible extensions of the work.
>
> > The main theoretical results hinge on the near-zero-sum smoothness parameter $\delta$ that measures how strongly coupled the game is. However, it remains unclear in which practical settings the $\delta$-smoothness assumption naturally holds or how to estimate it empirically. Although it is formally bounded by the overall smoothness $L$, it is uncertain how often $\delta$ would be sufficiently small in realistic problems to yield the claimed acceleration in convergence.
>
> **Please kindly check our general response on the estimation of $\delta$ for more detailed explanation.**
>
>
> > In addition, the presented results are developed specifically for two-player games, whereas the classical gradient complexity results for monotone general-sum games (e.g., Tseng, 1995; Nemirovski, 2004) extend to arbitrary numbers of players. It would strengthen the contribution to discuss whether the proposed framework and analysis can be generalized to multiplayer settings, or to explain the technical barriers that currently prevent such an extension.
>
> > Would you expect a similar analysis and complexity improvement to extend to the multiplayer setting? If not, could you elaborate on the main technical barriers that prevent such an extension? In particular, are there specific steps in the convergence proof or properties of the two-player potential function that fail to generalize when the number of players exceeds two?
>
> We believe that what has been extended to multiplayer games is the slow rate of $L / \min _i \mu_i$. However, the fast rate of $L / \sqrt {\mu \nu}$ has never been extended to multiplayer games to our knowledge.
>
> **Please kindly check our general response on extensions to multiplayer games.** We give both conceptual and technical reasons why the multiplayer games are structurally different, and the extension of the fast rate to three-player zero-sum games is already highly non-trivial.
>
> We thank the reviewer for suggesting the discussion of multiplayer games, and we updated our section 6 to include these discussions.
>
>
> > Moreover, could you provide additional examples or intuitions for practical game settings where the coupling part $g$ is expected to have a small smoothness parameter $\delta$? In other words, in what types of real-world or commonly studied games would the near-zero-sum assumption $\delta \ll L$ naturally hold?
>
> In the example of matrix games with transaction fee, the near-zero-sum parameter $\delta$ can be estimated from $\lVert \frac {A+B} {2} \rVert$; in the example of semi-cooperative games, the near-zero-sum parameter $\delta$ can be estimated from the smoothness parameter of the collaboration incentive.
>
> **Please kindly check our general response on the application examples.**  We have also revised our paper in section 4 and Appendix F to provide detailed scenarios, more economic backgrounds, and additional citations.
>
> We hope our response clarifies the applicability of near-zero-sum problem classes and the technical difficulty in extension to multiplayer games.

---

### Official Review · Reviewer_eGdp · 2025-11-01

**Soundness:** 3
**Presentation:** 3
**Contribution:** 2
**Rating:** 4
**Confidence:** 3

**Summary:**

The paper examines a new class of games that interpolates between zero-sum monotone games and general-sum monotone games. It is motivated by a gap between solving monotone zero-sum games and monotone general-sum games. The main contribution is an algorithm that improves the iteration complexity for that class. It finally provides some motivating examples together with some numerical results.

**Strengths:**

The paper is well written and examines a concrete problem. To my knowledge, the gap in the number of gradients needed between monotone zero-sum and monotone general-sum has been unexplored. This gap can be substantial, which motivates the main contribution of the paper. The paper also does a good job at explaining the technical steps and highlighting the technical contribution compared to existing techniques. The proposed algorithm is quite natural and clean; I didn't find any notable issues in the technical steps.

**Weaknesses:**

Overall, I believe the paper is lacking in motivating the significance of the new class of problems and providing good applications of it. Some of the examples discussed in Section 4 look pretty artificial. For example, this setting of matrix games with transaction fees hasn't been studied before, to my knowledge, and its significance is unclear. Concerning Example 2, combing competition with cooperation is, of course, very interesting. But the precise way it is formulated in that example appears to be again artificial. The paper would benefit a lot from having a good application that has been studied in the past and the community would appreciate. Otherwise, it is hard to see the significance of the paper.

The experimental section also doesn't go far enough. The only experiment is run on this setting of matrix games with transaction fees, and the improvement compared to OGDA only manifests itself for a narrow range of the parameter. The paper would benefit from expanding the set of experiments to include other domains where the method has an advantage.

**Questions:**

- In the experiments, can the authors also report the gap of the average iterate of OGDA instead of its last iterate? In theory that should converge faster. It would also be interesting to see whether alternating instead of simultaneous updates make a difference in that experiment.
- Can the authors provide more experiments using the class of competitive games with small cooperation incentives? Is there some specific setting in economics that naturally satisfies this assumption?

---

> ### Author Response · Authors · 2025-11-19
> **Response to Reviewer eGdp 1**
>
> We thank the reviewer for the feedback on our paper. The reviewer is satisfied with the theoretical motivation and technical contribution of our paper, while the reviewer also suggests including more application examples and adding more numerical experiments to the paper.
>
> > Overall, I believe the paper is lacking in motivating the significance of the new class of problems and providing good applications of it. Some of the examples discussed in Section 4 look pretty artificial. For example, this setting of matrix games with transaction fees hasn't been studied before, to my knowledge, and its significance is unclear. Concerning Example 2, combing competition with cooperation is, of course, very interesting. But the precise way it is formulated in that example appears to be again artificial. The paper would benefit a lot from having a good application that has been studied in the past and the community would appreciate. Otherwise, it is hard to see the significance of the paper.
>
> The main purpose of this work is to provide a new modelling of game scenarios to capture the near-zero-sum structure which has been missing in the existing models from literature. The applications we provided showed how this near-zero-sum structure naturally arises in practical game scenarios. The reviewer, however, is asking whether our new modelling has already been modelled in prior papers under this exact formulation. For sure, we are not expecting this to happen.
>
> **Please kindly check our general response on the application examples.**  We have also revised our paper in section 4 and Appendix F to provide detailed scenarios, more economic backgrounds, and additional citations.
>
> In brief, our practical novelty lies in the first to abstract the near-zero-sum structure. As shown in the application examples, this structure naturally arises in various games, but to our knowledge has not been properly modeled. Due to lack of studies, without this work, such games have *either* been **oversimplified as zero-sum games** *or* been **ignored and analyzed as general-sum games**. Our contribution, thus, bridges this practical gap.
>
> We respectfully disagree with the assessment that our examples are "artificial." Abstraction of game structure is a core methodological practice in the field. To illustrate this, we kindly refer the reviewer to a few important papers (all mentioned in the related work in Appendix A) and the examples therein. We think they are in a manner at least spiritually close to the examples provided in this work:
>
> - Example 1.3 in Ozan Candogan, Asuman Ozdaglar, and Pablo A Parrilo. Dynamics in near-potential games. Games and Economic Behavior, 82:66–90, 2013. (Introduced a relaxation of the strict Potential Games class.)
>
> - Section 2.1 in Kannan, Ravi, and Thorsten Theobald. "Games of fixed rank: A hierarchy of bimatrix games." Economic Theory 42.1 (2010): 157-173. (Introduced a low-rank structure in matrix games.)
>
> - Section 10 in Bharat Adsul, Jugal Garg, Ruta Mehta, Milind Sohoni, and Bernhard Von Stengel. Fast algorithms for rank-1 bimatrix games. Operations Research, 69(2):613–631, 2021. (Studied a rank-1 structure in matrix games.)
>
> These papers and examples demonstrate that abstracting game structure for computational gain is common practice in the field of game theory. Also, none of the new modelings in these papers have already been modelled in the exact form in the papers prior to them. Our introduction of the near-zero-sum class and our application examples are in this same spirit, and therefore, should not be considered artificial.
>
> > The experimental section also doesn't go far enough. The only experiment is run on this setting of matrix games with transaction fees, and the improvement compared to OGDA only manifests itself for a narrow range of the parameter. The paper would benefit from expanding the set of experiments to include other domains where the method has an advantage.
>
> > Can the authors provide more experiments using the class of competitive games with small cooperation incentives? Is there some specific setting in economics that naturally satisfies this assumption?
>
> **Please kindly check our general response on the basic numerical experiment.** We believe that our experiments are suitable to support the main theoretical claims. Regarding the economics background, we refer the reviewer to the [wikipedia page](https://en.wikipedia.org/wiki/Coopetition), and the remark by the end of Example 2 as well as the references therein.

---

> ### Author Response · Authors · 2025-11-19
> **Response to Reviewer eGdp 2**
>
> > It would also be interesting to see whether alternating instead of simultaneous updates make a difference in that experiment.
>
> We have compared our method with the two most classic methods, EG and OGDA. The experiment results clearly show the difference between our methods and these classic variational inequality methods, implying our method is the *only one* that successfully exploits the near-zero-sum structure. We believe that the basic numerical experiments provided in our paper are suitable to support the main theoretical claims of the paper.
>
> > In the experiments, can the authors also report the gap of the average iterate of OGDA instead of its last iterate? In theory that should converge faster.
>
> Why does the reviewer state that the average iterate should “converge faster”? *This statement is in fact incorrect.* At least in our experiment setting, both EG and OGDA have strong linear convergence rates for the last iterate as the function $h(\cdot, \cdot)$ is strongly convex-strongly concave [Tseng 1995; Theorem 4, Mokhtari et al. 2020].
>
> It is vital not to confuse this with the convex-concave cases, where the ergodic mean is typically required for the convergence rates.
>
> We hope the above clarifies all the issues raised in the reviews, and also highlights the application scenarios and experimental results.
>
> *Reference:*
>
> *Mokhtari, Aryan, Asuman Ozdaglar, and Sarath Pattathil. "A unified analysis of extra-gradient and optimistic gradient methods for saddle point problems: Proximal point approach." International Conference on Artificial Intelligence and Statistics. PMLR, 2020.*

---

> > ### Comment · Reviewer_eGdp · 2025-11-26
> >
> > I thank the authors for the detailed response. I will increase my score as the reviewers have partially addressed the concern regarding specific applications.

---

> > > ### Author Response · Authors · 2025-11-26
> > >
> > > Dear Reviewer eGdp: Thank you for taking the time to re-evaluate our submission and updating the score.

---

### Author Response · Authors · 2025-11-19
**General response 3**

### Estimation of parameter $\delta$:

To run our ICL algorithm, it is indeed necessary to know the $\delta$ parameter. To avoid confusion, we changed Assumption 3 to “Let $\delta > 0$ such that the function $g$ is $\delta$-smooth.”

In principle, one needs to know the $L$ parameter as well to run even the zero-sum algorithm, which can also be nontrivial to estimate in general. For monotone zero-sum games, we have not noticed any parameter-free algorithm that achieves the fast rate of $\frac L {\sqrt {\mu \nu}}$. This is indeed a common and non-trivial issue in many optimization problems.

We, however, argue that in practice we never meet pure black-box optimization [Nesterov 2005; Nesterov 2018]. When the games are given in closed form, the parameters of $\delta$ and $L$ can often be estimated. For instance, in section 5.1, we have explained how to compute the parameter $\delta$ from $\beta = \lVert \frac { A + B } {2} \rVert$ in regularized matrix games; and in section 5.2, the parameter $\delta$ corresponds to the smoothness of the collaboration incentive.

In the cases where it is, however, difficult to estimate these parameters, typical procedures in convex optimization are to perform line searches or to design parameter-free algorithms (like in the seminal paper of AdaGrad [Duchi et al. 2011]). Since for zero-sum cases such extensions are still open, we leave the further extensions of our near-zero-sum algorithm for future work.

*References:*

*Von Neumann, John, and Oskar Morgenstern. "Theory of games and economic behavior." Theory of games and economic behavior. Princeton university press, 1947.*

*Nesterov, Yu. "Smooth minimization of non-smooth functions." Mathematical programming 103.1 (2005): 127-152.*

*Nowak, Martin A. Evolutionary dynamics: exploring the equations of life. Harvard university press, 2006.*

*Duchi, John, Elad Hazan, and Yoram Singer. "Adaptive subgradient methods for online learning and stochastic optimization." Journal of machine learning research 12.7 (2011).*

*Nesterov, Yurii. Lectures on convex optimization. Vol. 137. Berlin: Springer International Publishing, 2018.*

---

### Author Response · Authors · 2025-11-19
**General response 2**

### Basic numerical experiment:

Our experimental section is designed as a *basic numerical validation* of our main theoretical findings (which is our paper's primary focus). We chose the regularized matrix game as our test bed due to its clean structure and amenability to generate random examples at a relative scale. Our basic numerical experiments successfully validate our core claims:

1. Our proposed ICL algorithm converges faster as the game approaches the zero-sum structure ($\delta \to 0$);

2. Classical variational inequality methods (EG/OGDA) do not benefit from the near-zero-sum structure; and

3. Consequently, our ICL algorithm is faster than the classic methods when the game is sufficiently near-zero-sum.

We believe that in the paper we already have suitable numerical results to support the main theoretical claims of the paper.

Reviewer AbM8 gives more detailed feedback on the convergence plot of Figure 1. While all 21 entries (seven entries for each of the three algorithms) are plotted, the perceived "lack of distinct lines" for EG and OGDA is, in fact, *a crucial empirical result consistent with our theory*.

The seven lines for EG (Green) and OGDA (Red) almost entirely merge because the convergence rate of classical variational inequality methods is determined by $\frac{L} {\min \\{\mu, \nu \\}\}$, and cannot exploit the near-zero-sum parameter $\delta$. Conversely, the seven lines for our proposed Black-Box Reduction Algorithm (Blue) are visibly separated, demonstrating its dependence on the $\delta$ parameter and empirically validating its ability to harness the near-zero-sum structure for faster convergence.

To make this point clearer for the reader, **we have updated the description of experiment results (section G.2)** to explicitly state that the overlap of the OGDA and EG lines is an expected empirical result consistent with existing theory.

### Application examples:

The examples provided in our paper serve as canonical abstractions showing how the near-zero-sum structure naturally arises in games:

- For instance, *matrix games with transaction fees* are analogous to (a) betting sites often set odds such that there is overall no loss-gain (zero-sum game) but charge a small transaction fee; or (b) negotiation between two clients via a trusted-third party that charges a brokerage fee is another example of a case with small transaction fee.

- Similarly, we have already presented a lot of citations for games where *competition and cooperation co-exists* (see the list of references at the end of Example 2). Our work studies the parameter range where the cooperation incentive is small. For example, repeated Prisoners' dilemma with a stochastic number of rounds with small benefit-to-cost ratio is an example of competition with small cooperation incentive (as benefit-to-cost ratio $b/c$ is small) [Chapter 8.7, Nowak 2006].

This paper is the first formal introduction of a new class that precisely models the near-zero-sumness often ignored in existing literature—for instance, the impact of third-party effects or small external factors:

- Before our work, we were not aware of any study of near-zero-sum games, hence existing modellings were forced to choose between *either* the *inaccurate, perfectly zero-sum simplification* *or* the *computationally costly general-sum model*;

- Now, with our work, we provide a *mathematically elegant framework* to analyze the missing issue in these real-world competitive scenarios.

Hence, the applications provided in the paper illustrate that the near-zero-sum structure is a general feature that our algorithm is the first to efficiently exploit.
**In our revision, we added these additional explanations in section 4 and the more concrete illustrations in Appendix F.** We believe that the additional backgrounds and references will be helpful for the readers outside the field of game theory.

---

### Author Response · Authors · 2025-11-19
**General response 1**

We thank all the reviewers for their insightful feedback.

The contributions of this paper are threefold:

1. the motivation and formal introduction of the new class of monotone near-zero-sum games;

2. the design of a general black-box reduction algorithm with faster convergence rate; and

3. the demonstration of the new problem class’s applicability through practical, literature-motivated game scenarios.

Our paper provides the first core and general analysis for two-player monotone *non-zero-sum* games with *general conditioning*.
**All the reviewers *unanimously recognize* the technical contribution of our paper.**

> Reviewer eGdp: The paper also does a good job at explaining the technical steps and highlighting the technical contribution compared to existing techniques.

> Reviewer F4W2: The problem is theoretically well-motivated. The proposed Iterative Coupling Linearization (ICL) algorithm is conceptually novel and algorithmically elegant, and the convergence analysis is technically rigorous.

> Reviewer AbM8: On a conceptual level, the paper does take a step in filling the gap between zero sum and general sum games, introducing an interesting class. Moreover, from the technical side, the proofs require technical work.

> Reviewer BPVS: The research question studied is of clear interest to the community, and is well-motivated in the introduction. The use of the potential function and linearization in the design of ICL is a nice idea, and leads to improved/optimal convergence rates in certain game classes. The technical components of the paper are clean and easy to digest.

While our work makes the first core analysis towards faster rates for near-zero-sum games, as noted in our paper and as mentioned by the reviewers, there are many non-trivial and exciting avenues for future research. Indeed, many of them (e.g., 4 & 6) are still open for the (basic) zero-sum games:

4. The removal of the additional logarithmic factor from the current upper bound, and the extension to adaptive/line-search algorithms;

5. The derivation of tighter lower bounds for the non-zero-sum classes;

6. The generalization to multiplayer (near-zero-sum) settings;

7. Additional experiments on more or larger examples;

……


In our general response, we address some common issues raised by the reviewers.

### Extensions to multiplayer games:

Our work focuses on achieving fast convergence rates for two-player monotone non-zero-sum games, a crucial first generalization of the two-player monotone zero-sum games. A natural inquiry raised by the reviewers is whether these fast rates can be extended to multiplayer near-zero-sum games.

We emphasize that this generalization would require solving two highly non-trivial challenges: (a) extending the previous results of the fast rate from the two-player monotone zero-sum case to the multiplayer monotone "zero-sum" case, and (b) subsequently generalizing the resulting multiplayer "zero-sum" framework to the "near-zero-sum" class.

Given that the default rate for an $n$-player monotone game is slow, $\mathcal{O}(L / \min_i { \mu_i })$, the extension of the previous results in step (a) is already a challenging open problem. Thus, the full extension of our result to multiplayer games is a major open problem that we leave for future work.

Reviewers F4W2 and BPVS ask for technical reasons why we did not generalize the faster rates to multiplayer games. We are happy to share more insights in this regard. The primary difficulty lies even in the first step (the multiplayer zero-sum case), which is largely unexplored:

- Conceptually, the zero-sum assumption in two-player games implies strict competition, which is not guaranteed to hold between any two players in a general multiplayer zero-sum game.

- More technically, classic results (e.g., von Neumann & Morgenstern, 1947) show that an $n$-player general-sum game can be reduced to an $(n+1)$-player zero-sum game by letting the $(n+1)$th player take the negative of the summation of the first $n$ players. This implies that three-player zero-sum games are inherently no easier than two-player general-sum games.

The above discussions suggest that achieving a fast convergence rate in the multiplayer setting will likely require significantly stronger structural assumptions than the simple summation of utilities to zero.
Given such complexity, a comprehensive treatment of multiplayer games is beyond the scope of this paper.

**We have included the above discussions in the future work paragraph.**

---

### Author Response · Authors · 2025-11-19

Dear reviewers,

Thank you for your helpful feedback!

We have given a general response (1/2/3) for the common issues raised by the reviewers, and more detailed responses for each reviewer. We have also updated our paper accordingly.

Please kindly let us know if we have addressed your concerns.
Thanks!

**=====Remark to ACs after the reversion to original reviews=====**

All reviewers recognized the novelty and theoretical contribution of our work. Prior to the score rollback, **the only initially negative reviewer (eGdp) had already raised their score to positive**, so all reviewers were supportive.

In paper revision, with major updates highlighted in blue (pages 8, 9, 10, 22, 23), we have *added background on applications, discussed technical extensions, provided concrete examples, and clarified the experimental results*. We hope these improvements address the reviewers’ concerns.

-Authors

---

### Meta-Review · Area_Chair_nV9V · 2026-01-12

**Summary:**

The paper introduces a new class of two-player \emph{monotone} (concave, not necessarily normal form) games called near zero-sum. This class of games is associated with a parameter $\delta$ that indicates the smoothness of the sum of the utility functions, so if $\delta$ is zero, the underlying game is zero-sum otherwise general-sum and how small $\delta$ is captures the proximity to a zero-sum game. A black-box reduction/algorithm is provided between this class of games and monotone two-player zero-sum games and as a result an algorithm that finds an \epsilon-approximate NE in the same number of gradient calls as the best known algorithms for monotone zero-sum games. Moreover, experiments are provided.

**Reviewer Concerns:**

Most of the reviewers were positive about this work. One reviewer raised some concerns about the experimental section and the importance of the class of games defined in this paper. It seems that the authors addressed some of the concerns during the rebuttal by improving the experimental section. The AC believes that the paper though interesting, has failed to explain the significance of the new defined class of games. Moreover, the class of games has unique equilibria (or if just monotone, unique payoff) which makes it less appealing as a setting, even if it is hard to analyze two-player general-sum (non-monotone) games. Overall the AC is not very enthusiastic about this work and believes it is a borderline paper. Nevertheless, the AC recommends acceptance due to the high review scores.

**Reviewer Scores:**

The authors that was negative showed interest in increasing his/her score. The other reviewers were positive.

---

### Decision · Program_Chairs · 2026-01-26

Accept (Poster)